# Racial and ethnic differentials in COVID-19-related job exposures by occupational standing in the US

Noreen Goldman[1]*, Anne R. Pebley[2], Keunbok Lee[3], Theresa Andrasfay[4], Boriana Pratt[5]

1 Office of Population Research, School of Public and International Affairs, Princeton University, Princeton, New Jersey, United States of America, 2 California Center for Population Research, Fielding School of Public Health, University of California Los Angeles, Los Angeles, California, United States of America, 3 California Center for Population Research, University of California Los Angeles, Los Angeles, California, United States of America, 4 Leonard Davis School of Gerontology, University of Southern California, Los Angeles, California, United States of America, 5 Office of Population Research, Princeton University, Princeton, New Jersey, United States of America

* ngoldman@princeton.edu

**Data Availability Statement:** All data used in this analysis are publicly available. IPUMS USA Version 10.0 is available at https://doi.org/10.18128/D010.

## Abstract

Researchers and journalists have argued that work-related factors may be partly responsible for disproportionate COVID-19 infection and death rates among vulnerable groups. We evaluate these issues by describing racial and ethnic differences in the likelihood of work-related exposure to COVID-19. We extend previous studies by considering 12 racial and ethnic groups and five types of potential occupational exposure to the virus: exposure to infection, physical proximity to others, face-to-face discussions, interactions with external customers and the public, and working indoors. Most importantly, we stratify our results by occupational standing, defined as the proportion of workers within each occupation with at least some college education. This measure serves as a proxy for whether workplaces and workers employ COVID-19-related risk reduction strategies. We use the 2018 American Community Survey to identify recent workers by occupation, and link 409 occupations to information on work context from the Occupational Information Network to identify potential COVID-related risk factors. We then examine the racial/ethnic distribution of all frontline workers and frontline workers at highest potential risk of COVID-19, by occupational standing and by sex. The results indicate that, contrary to expectation, White frontline workers are often overrepresented in high-risk jobs while Black and Latino frontline workers are generally underrepresented in these jobs. However, disaggregation of the results by occupational standing shows that, in contrast to Whites and several Asian groups, Latino and Black frontline workers are overrepresented in lower standing occupations overall and in lower standing occupations associated with high risk, and thus may be less likely to have adequate COVID-19 protections. Our findings suggest that greater work exposures likely contribute to a higher prevalence of COVID-19 among Latino and Black adults and underscore the need for measures to reduce potential exposure for workers in low standing occupations and for the development of programs outside the workplace.

V10.0 and O*NET data are available at https://www.onetcenter.org/db_releases.html.

**Funding:** Research reported in this publication was supported by the National Institute on Aging through grant numbers R01AG061094 (UCLA & Princeton, ARP and NG) and T32AG000037 (TA) and the Eunice Kennedy Shriver National Institute on Child Health and Human Development through grant number P2CHD041022 (ARP). The funders had no role in study design, data collection and analysis, decision to publish or preparation of the manuscript.

**Competing interests:** The authors have declared that no competing interests exist.

## Introduction

In the United States, Black, Latino, and Native American adults have experienced substantially higher rates of COVID-19 infection and mortality during 2020 than Whites and Asians [1–7]. Researchers and journalists argue that the differences are due, at least in part, to two work-related factors: (1) Black, Latino, and Native American workers are more likely to hold jobs that have to be done at their workplace rather than remotely, and (2) Latino and Black workers face greater risks of exposure to COVID-19 in their jobs than others [3, 8–18]. This argument is consistent with the long history and contemporary effects of structural racism on occupational segregation in the US [19–22]. In this paper, we investigate differences in the likelihood of work-related exposure to COVID-19 by race and ethnicity. Our goal is descriptive: we provide insights into the pandemic by presenting the size and scope of these racial and ethnic disparities among workers rather than by estimating causal models of their determinants.

It is difficult to quantify the importance of occupational vs. other exposures to the coronavirus. Most studies have used two approaches to argue that occupational risks to COVID-19 are higher for marginalized racial and ethnic groups than for Whites. We use a combination of these approaches. The most common strategy has been to look at racial/ethnic differences in recent (generally pre-pandemic) employment in industries or occupations that are considered essential or frontline during the pandemic [8, 23, 24]. As we discuss later, these earlier studies have been based on various definitions of essential and frontline. A second approach, sometimes used in conjunction with the first, has been to estimate relative risk of different groups during the pandemic from survey data about whether jobs entail high exposure to disease or infection and/or require close proximity to other people [9]. Based on the recent availability of death records for 2020 in some locations, several studies have employed an additional strategy. They have used occupational codes from death certificates to directly identify occupations with exceptionally high mortality during the pandemic period and examine differential mortality by race and ethnicity [18, 25].

We extend this work in several ways. First, we consider 12 racial/ethnic groups. Previous studies on occupations and COVID-19 have largely been limited to four broad racial/ethnic categories (White, Black, Latino and Asian), although workers in other racial/ethnic groups or in Asian and Latino subgroups may be disproportionately exposed to COVID-19 [6]. Second, we examine five distinct job characteristics that could expose individuals to a high risk of contracting COVID-19: exposure to disease/infection; proximity to others; face-to-face discussions; interactions with external customers or the public; and working mostly indoors. We also present separate results by sex because of vastly different occupational profiles for men and women.

Third, we examine potential risk of exposure separately by occupational standing (OS), defined here as the proportion of workers in each occupation with at least some college education. We use OS as a proxy for workers' access to COVID-19 mitigation measures in the workplace. As health inequality research shows, those with higher socioeconomic status (e.g., OS, education or income) generally have access to a wider array of resources, including power and influence, to protect health than others do [26]. In the case of potential workplace exposure to COVID-19, higher OS workers are more likely: (a) to work for employers who voluntarily practice risk mitigation and provide PPE and other tools for workers to do so, (b) if necessary, to demand risk reduction measures and to have the bargaining power to obtain them, and (c) to understand (or learn about) COVID-19 transmission routes and to comply with risk reduction strategies or implement their own [26–28]. Occupational standing is closely linked to having greater control over working conditions [29–31]. For example, despite the high risks often faced by physicians, nurses and other health personnel at the start of the pandemic when PPE

was not readily available, these higher OS workers are generally more likely to work in environments that take disease transmission risk mitigation seriously and they have more power and knowledge to insist on effective risk mitigation when it is not available. Employers of lower OS workers in meat processing factories, farms, and retail stores, for example, are less likely to provide risk mitigation and workers have less power and knowledge to demand appropriate measures. Perhaps it is no surprise that lower OS workers appear to face especially high exposure to COVID-19 in the workplace [17, 27].

Examining results by occupational standing leads to a clearer picture of racial/ethnic differentials in potential exposure to COVID-19 transmission. Our results best reflect the situation at the end of 2020 when this paper was originally submitted. Although the situation has changed with widespread vaccination, lower OS and Black and Latino populations and those who are socioeconomically disadvantaged are less likely to be fully vaccinated than others [32], so that the picture we describe remains, unfortunately, relevant.

## Data

We use data from two sources. Information on employment status, type of current/most recent job or business during the past five years, years of education, sex and race/ethnicity come from the 2018 American Community Survey (ACS), which was the most recent large, nationally representative survey fielded prior to the pandemic and available at the time of this research. Access is provided by the Integrated Public Use Microdata Series (IPUMS) [33]. Variables related to work context are drawn from the Occupational Information Network (O*NET), a database of work characteristics by detailed occupation collected by the US Department of Labor [34].

O*NET obtains information about job characteristics–such as work-related activities, work environment, and skills required to do the job–for almost 1000 detailed occupations. We use data from O*NET version 24.3, which contains information collected through 2019 and was released in May 2020 [34]. We identified six variables from O*NET surveys of incumbent workers that capture hazardous job characteristics that may expose workers to SARS-CoV-2 [17, 27]–which appears to be transmitted primarily through aerosols and respiratory droplets when an infected person coughs, sneezes, talks or breathes near others [35]. The six workplace variables reflect five types of risk: one variable for each of exposure to infection, physical proximity to others, face-to-face discussions, and interactions with external customers and the public, and two variables that reflect working indoors. Note that, like other studies, we do not have a direct indicator of actual exposure to the coronavirus, nor can we observe COVID-19 infections among workers.

Approval for this research was obtained from the Institutional Review Boards at the University of California Los Angeles and Princeton University.

## Variables

**Race/ethnicity.**   Individuals were classified according to the following 12 race/ethnicity categories as reported by the head of the household in the ACS: White, Black, Latino, Chinese, Filipino, Vietnamese, Korean, other Asian, Native American (American Indian/Alaskan Native), Pacific Islander (Native Hawaiian/Pacific Islanders), mixed race, and other race/ethnicity. The three largest groups in the other Asian category are Indian, Japanese and Pakistani. In our analysis, all racial/ethnic categories except Latino exclude those who reported themselves as Latino/Hispanic in the ACS question on ethnicity.

**Occupational standing.**   Neither our data nor any other nationally representative data that we are aware of contain information on the use of COVID-19 mitigation measures. In order to rank occupations, we use a well-established sociological indicator of occupational

standing, known as "occupational education" [36] as a proxy for access to workplace risk mitigation measures. Occupational standing is measured at the occupation, not the individual, level. For each occupation, we calculate occupational education from the ACS as the percent of all workers in the occupation who completed at least one year of college education (note that this variable classifies individuals by the educational attainment of *all workers* in their occupation, *not* their own educational attainment). We use this measure of occupational standing or "OS" for simplicity, to divide the 409 occupations reported by ACS respondents in our analysis into quartiles, i.e., each group comprising about one-quarter of the occupations (not one-quarter of the workers). We chose to use occupational education rather than occupational income–another well-established indicator of occupational ranking–both because educational attainment is typically reported with greater accuracy than income or wages and because among those with a valid occupational code in the ACS data file provided by IPUMS, educational attainment information is complete while information on income is missing for one-fifth of individuals. The correlation between these two measures of OS across the occupations in the dataset is high (0.72), indicating that occupational income and occupational education would produce similar rankings of occupational standing (OS).

**Frontline status.**   During the pandemic, many workers have been sheltered from occupational exposure to infection by working remotely, e.g., from home. To distinguish between workers who are more vs. less able to work remotely, studies have used varying criteria to define "essential" and/or "frontline occupations." For example, some studies have used guidelines issued by the Department of Homeland Security (DHS) Cybersecurity and Infrastructure Security Agency (CISA) to define essential occupations or industries [9, 23, 24, 37], while others have defined a set of occupational categories without providing specific selection criteria [38]. In this analysis, we include only occupations classified as "frontline" based on the definition offered by Dingel and Neiman [39] and used by Blau et al. [24]: occupations in which one-third or fewer workers can feasibly work from home, ascertained from responses to 15 questions in O*NET. A total of 249 out of 409 occupations in our analysis are considered frontline according to these criteria.

We do not consider whether occupations are classified as essential or subject to lockdown because there has been enormous variation across states, localities, and time period in definitions and their application during the pandemic. For example, janitors, maids, bus drivers, retail sales workers, and personal care workers would not be classified as essential according to industry guidelines issued by CISA [24], yet many individuals in these occupations have likely been working away from home much or most of the time since March, 2020.

**Defining high risk occupations.**   The five types of risk we consider from O*NET data–exposure to infection, physical proximity to others, face-to-face discussions, interactions with external customers and the public, and working indoors–are based on the following O*NET questions (in their original wording):

1. How often does your current job require that you be exposed to diseases or infection?

2. How physically close to other people are you when you perform your current job?

3. How often does your current job require face-to-face discussions with individuals and within teams?

4. In your current job, how important are interactions that require you to deal with external customers (as in retail sales) or the public in general (as in police work)?

5. How often does your current job require you to work outdoors, exposed to all weather conditions? [And:] How often does your current job require you to work outdoors, under cover (like in an open shed)?

Each of these questions has five possible categorical responses; the responses reflect frequency of exposure for questions 1,3, and 5; degrees of closeness for question 2, and importance of interactions for question 4.

For each of the first four risk indicators, we define high risk occupations as those for which the mean response in the O*NET data falls in the *highest* quartile (25%) of the full set of 409 occupations in the analysis. For the fifth risk indicator we use the two questions shown above reflecting the frequency with which employees work outdoors; the responses for these two questions are highly correlated. Because indoor work is higher risk than outdoor work *ceteris paribus*, we take the maximum of the O*NET values for these two variables and define high risk as the quartile of occupations with the *lowest* values, thereby identifying jobs with the lowest frequency of working outdoors as high risk.

## Analytic strategy

In order to match occupations reported in the ACS with characteristics in O*NET, we converted the occupation code (Standard Occupational Classification or SOC) in O*NET into the four-digit 2010 census occupational code recorded in the 2018 ACS using a crosswalk provided at https://www.bls.gov/emp/documentation/crosswalks.htm. In cases where one census occupational code corresponded to multiple O*NET occupations, we took unweighted averages of the characteristics of the O*NET occupations and assigned these to the census occupational code. We linked occupations between the ACS and O*NET for all employed persons who reported holding a job in the ACS with the exception of those in the military (O*NET data were not available for military occupations, which were held by about 6000 individuals in the ACS); this linkage yielded about 1.9 million individuals in the ACS in 409 occupations. Approximately 88 percent of these individuals reported about the job they held during the past year with the remaining respondents reporting about the job held between one and five years prior to the survey.

It is important to note that unemployment and labor force participation rates vary considerably by race, ethnicity, and sex. For example, in the third quarter of 2020, the unemployment rate for Black men 16+ years old was 13.8 percent compared to 7.4 percent for White men and 9.6 percent for Asian men [40]. Moreover, job losses during the first half of 2020 occurred disproportionately among Black, Latino, and Asian workers and were especially likely to be experienced by those whose jobs could not be performed at home, most notably frontline workers with little job security and low pay [41, 42]. Another problem with obtaining an accurate description of the labor force is that unemployment rates have fluctuated throughout the period of the pandemic. As a consequence of these issues, our analysis does not account for differential unemployment and labor force participation rates.

As noted above, the analysis includes only individuals who worked in the five years prior to the ACS, the vast majority of whom worked in the year prior to the ACS; we refer to these individuals as either "recent workers" or simply as "workers." After determining the percent of workers in each racial/ethnic group that hold frontline occupations, we measure the overrepresentation vs. underrepresentation of frontline workers in high-risk occupations. For each of the five risk factors, these estimates denote the proportion of frontline workers in the highest quartile of exposure that are in a given racial/ethnic group *relative to* the proportion of all frontline workers that are in that racial/ethnic group. We subsequently stratify workers in the highest risk group according to their quartile of our OS measure. The first OS quartile comprises occupations with the lowest standing. Similar to the unstratified estimates, we then consider the proportion of workers in a specified OS quartile (for each of the five risk factors) that are in a given racial/ethnic group relative to the proportion of all frontline workers that are in that group

The descriptive results are presented in five figures: the first includes all 12 racial/ethnic groups, whereas each of the remaining graphs, which are stratified by OS, considers three racial/ethnic groups. All of the figures are heatmaps, with shades of orange indicating overrepresentation of workers in a given risk category relative to all frontline workers (for the particular racial/ethnic group), and shades of green indicating the corresponding underrepresentation in a given risk category. Progressively darker shades reflect successively more extreme values of overrepresentation (orange) or underrepresentation (green). The figures are based on separate calculations for male and female frontline workers.

All calculations use weights provided in the ACS. STATA/MP Version 15.1 was used for the analysis and production of figures [43]. The data underlying these graphs are presented in S1 Appendix.

## Results

Table 1 shows the percentages of workers in frontline occupations by sex and race/ethnicity. Employment in frontline occupations varies considerably across groups. For both men and women, Latinos are the most likely workers to hold frontline occupations whereas Chinese and Korean workers are the least likely to do so. Latino, Black, Native American, and Pacific Islander men are the most likely to have frontline occupations–more than 70% of male workers in each of these groups are classified as frontline. In contrast, Vietnamese, Latino, and Filipino women are the most likely female workers to hold frontline occupations. Asian women are at least as likely as Asian men to work in frontline occupations.

Fig 1 depicts overrepresentation and underrepresentation of the 12 racial/ethnic groups for each of the five occupational exposures to potential viral transmission, by sex. For example, consider the value of 0.80 for Latino males regarding potential risk of infection (top row, third column of Fig 1). This value suggests that Latinos are underrepresented by 20% in frontline jobs that entail a high risk of infection relative to the proportion of frontline workers that are Latino. It can be calculated from S1 Appendix in the following way: Latino males comprise 17.4% of frontline jobs in the highest quartile of infection risk (5th column of S1.4) and 21.6%

**Table 1. Number of recent workers and percentage in frontline occupations by race/ethnicity and sex.**

| Race/Ethnicity | Male | | Female | |
|---|---|---|---|---|
| | Number of Recent Workers | Frontline (%) | Number of Recent Workers | Frontline (%) |
| White | 675,542 | 59.5 | 632,421 | 46.2 |
| Black | 76,649 | 72.9 | 87,168 | 57.5 |
| Latino | 133,868 | 77.9 | 120,853 | 61.8 |
| Chinese | 13,095 | 40.9 | 14,265 | 41.3 |
| Filipino | 7,798 | 61.7 | 10,403 | 61.2 |
| Vietnamese | 5,150 | 63.5 | 5,519 | 66.2 |
| Korean | 4,303 | 44.0 | 4,684 | 47.8 |
| Other Asian | 22,072 | 42.1 | 18,654 | 57.9 |
| Native American | 7,018 | 75.5 | 7,059 | 56.7 |
| Native Hawaiian/Pacific Islander | 1,402 | 73.0 | 1,353 | 54.1 |
| Mixed Race | 17,555 | 63.2 | 17,857 | 52.0 |
| Other Race | 1,695 | 64.3 | 1,778 | 57.3 |
| Total | 966,147 | 63.7 | 922,014 | 50.5 |

Note: Data are from the 2018 American Community Survey (ACS). Percentages are calculated using weights provided by the ACS; numbers are unweighted counts of ACS respondents. All racial/ethnic categories except Latino exclude those who reported themselves as Latino/Hispanic in the ACS question on ethnicity.

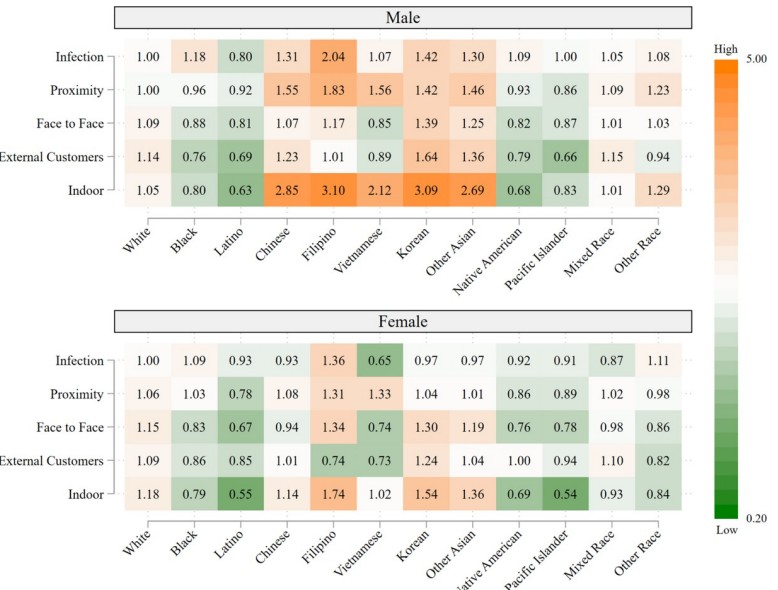

**Fig 1. Representation of frontline workers in high-risk occupations relative to representation as frontline workers, by race/ethnicity and sex.** Data are from the 2018 American Community Survey (ACS). Occupations are considered to be high-risk if the average value for the risk factor in the O*NET data falls in the highest quartile of the occupations in the analysis.

of all frontline workers (5th column of S1.2); 17.4/21.6 = 0.806 or, allowing for slight round-off error, the value of 0.80 in Fig 1.

Despite elevated COVID-19 mortality among Latino, Black, and Native American populations, Fig 1 shows that these workers are generally *underrepresented* in high-risk occupations; the main exception is a modest overrepresentation of Black workers in occupations involving risk of infection. In contrast, Whites are modestly overrepresented according to several components of risk, particularly exposure to external customers for men and indoor work for women. The most extreme results are those for Asians, with all ethnic groups of Asian men having more than a twofold overrepresentation in occupations involving indoor work; Filipino men are additionally highly overrepresented in occupations involving high risk of infection.

Fig 1 highlights the complexity of identifying the racial/ethnic groups with the highest potential occupational exposure to the virus. Contrary to expectation based on earlier studies and the media, the values in Fig 1 would suggest that Black and Latino workers do not face higher occupational risks related to viral exposure compared with Whites. However, this conclusion is likely to be incorrect. The classification of high-risk occupations that we have considered thus far does not capture likely variation among occupations in COVID-19 risk reduction measures that are deployed in the workplace and are more likely to be implemented in higher OS occupations. For example, home health aides (relatively low OS and disproportionately held by workers of color) and physicians (high OS and disproportionately White) are both frontline occupations at high risk for infection and physical proximity to other people, but physicians are far more likely to have access to and use PPE, work in frequently sanitized environments, and have had specialized training to reduce potential exposure to infections at work [44, 45].

To get a clearer picture of COVID-19 transmission-related risks faced by workers in different occupations and different racial/ethnic groups, the remaining graphs use OS quartile as a proxy measure for workplace risk mitigation during the COVID-19 pandemic. As discussed

above, our use of OS as a proxy measure is based on the assumption that workers in higher OS occupations are more likely to have access to PPE and other mitigation measures compared to workers in lower OS occupations. The most common frontline occupations in the highest OS (4th) quartile are predominantly in the healthcare sector, most notably registered nurses, physicians, and surgeons. In contrast, the most common frontline occupations in the lowest OS (1st) quartile include cashiers, drivers, janitors, laborers, stock clerks, and housekeeping cleaners. S2 Appendix lists the most frequently held frontline occupations for each of the five risk indicators, stratified by OS quartile. Only occupations that include at least five percent of workers in the quartile are shown. Although many of the high-risk occupations across all quartiles are in the healthcare sector, numerous occupations outside of the healthcare sector are high-risk on at least one of the five indicators. In the lowest OS quartile, these high-risk occupations include janitors, maids, and other cleaners; carpenters; maintenance and repair workers; food preparation workers and servers; machine operators; stock clerks; and cashiers.

Fig 2, which stratifies the estimates in Fig 1 by occupational standing for White, Black, and Latino workers, provides greater insight into racial/ethnic variation in occupational risks than Fig 1. Among workers in high-risk occupations on each of the five indicators, White workers, both men and women, are generally overrepresented in the 3rd and especially the 4th (i.e., highest) OS quartile. Black and Latino workers, on the other hand, are generally overrepresented in the 1st quartile while being vastly underrepresented (especially Latinos) in the 4th OS quartile. Latino male workers have the largest overrepresentation (1.5 or 50% greater representation relative to their proportion of all frontline workers) in the lowest OS quartile associated with physical proximity, working, for example, as carpenters, roofers, and food preparation and serving-related workers. Latino female workers have the largest overrepresentation (2.1) in the lowest OS occupations associated with high infection risk, working, for example, as maids and other cleaners.

Figs 3–5 present heatmaps for the remaining racial/ethnic groups, in the same format as Fig 2. As shown in Fig 3, Chinese, Filipino and Vietnamese men, as well as Filipino women, occupy a much larger share of high-risk, but also high OS, occupations relative to their share of frontline occupations, often three to four times as large; this occurs across all five risk

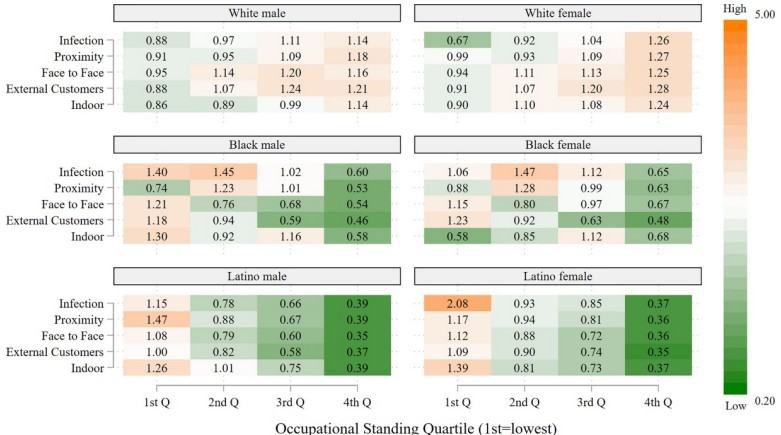

**Fig 2. Representation of frontline workers in high-risk occupations relative to representation as frontline workers by occupational standing: White, Black and Latino workers, by sex.** Data are from the 2018 American Community Survey (ACS). Occupations are considered to be high-risk if the average value for the risk factor in the O*NET data falls in the highest quartile of the occupations in the analysis. Occupational standing (OS) is defined as the percentage of ACS respondents reporting this occupation who completed at least one year of college education. The 1st OS quartile is the lowest and the 4th OS quartile the highest.

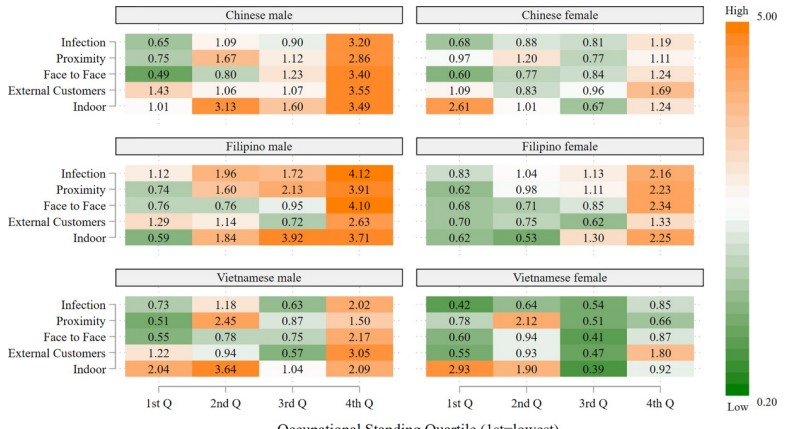

**Fig 3. Representation of frontline workers in high-risk occupations relative to representation as frontline workers by occupational standing: Chinese, Filipino, and Vietnamese workers, by sex.** Data are from the 2018 American Community Survey (ACS). Occupations are considered to be high-risk if the average value for the risk factor in the O*NET data falls in the highest quartile of the occupations in the analysis. Occupational standing (OS) is defined as the percentage of ACS respondents reporting this occupation who completed at least one year of college education. The 1st OS quartile is the lowest and the 4th OS quartile the highest.

indicators. In general, the overrepresentation of men in high-risk jobs with high OS exceeds that for women in these ethnic groups. As shown in the top two panels of Fig 4, a similar pattern occurs for Korean and other Asian workers, i.e., large overrepresentation, especially for men, in high-risk, high OS occupations. The majority of Filipino men and women, and Chinese and Korean women, who are in high OS frontline occupations are registered nurses. The other frequently held occupations for these groups are also in the healthcare sector. Chinese and Korean men with the highest OS frontline occupations are most likely to be physicians and surgeons, but are also employed as dentists, pharmacists and registered nurses. The high proportion of Asians in the healthcare sector largely accounts for their disproportionate

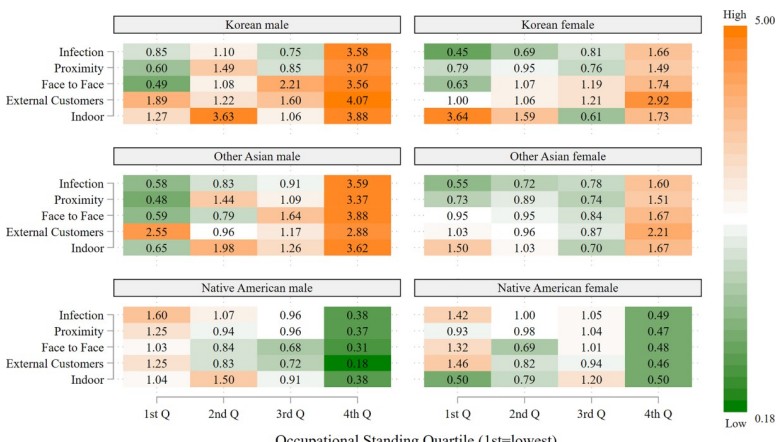

**Fig 4. Representation of frontline workers in high-risk occupations relative to representation as frontline workers by occupational standing: Korean, other Asian, and native American workers, by sex.** Data are from the 2018 American Community Survey (ACS). Occupations are considered to be high-risk if the average value for the risk factor in the O*NET data falls in the highest quartile of the occupations in the analysis. Occupational standing (OS) is defined as the percentage of ACS respondents reporting this occupation who completed at least one year of college education. The 1st OS quartile is the lowest and the 4th OS quartile the highest.

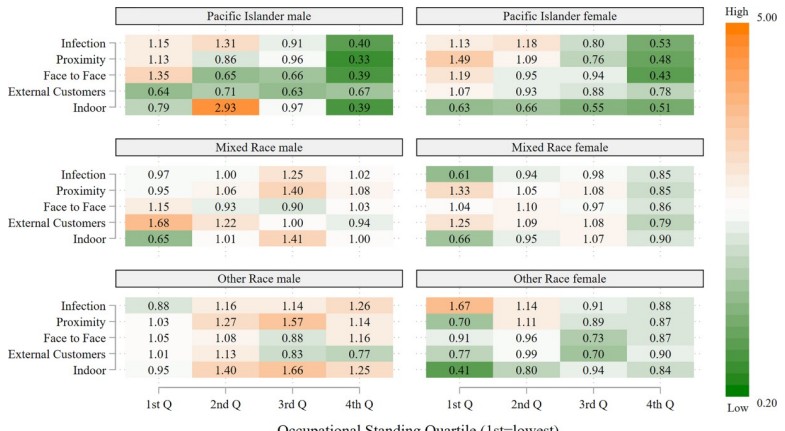

**Fig 5. Representation of frontline workers in high-risk occupations relative to representation as frontline workers by occupational standing: Pacific Islander, mixed race, and other race workers, by sex.** Data are from the 2018 American Community Survey (ACS). Occupations are considered to be high-risk if the average value for the risk factor in the O*NET data falls in the highest quartile of the occupations in the analysis. Occupational standing (OS) is defined as the percentage of ACS respondents reporting this occupation who completed at least one year of college education. The 1st OS quartile is the lowest quartile and the 4th (OS) quartile the highest.

presence in high-risk occupations, particularly those involving exposure to infection, close proximity to others, and indoor locations.

These findings for Asian workers contrast strikingly with those for Native Americans in the bottom panel of Fig 4. Native American workers, both men and women, occupy high-risk jobs of the lowest OS quartile more frequently than expected relative to their numbers as frontline workers, particularly for jobs that entail a high risk of infection. Patterns for Pacific Islanders, shown in the top panel of Fig 5, are similar to those of Native Americans. No clear pattern emerges for the remaining heterogeneous racial/ethnic groups considered, mixed race and other race.

## Discussion

In this study, we extended earlier work on racial/ethnic differences in potential work-related exposure to COVID-19 by examining 12 racial/ethnic groups and by considering five indicators of potential risk exposure. The analysis is descriptive–portraying potential workplace risks by race, ethnicity and gender–rather than an investigation of the root causes of these patterns. Our central innovation is to use occupational standing (OS) as a proxy for whether workers and their workplaces employ significant COVID-19-related risk reduction strategies (PPE, distancing, sanitation, improved ventilation, etc.)–something not measured in the data we use nor in any other large, nationally representative data sets, to our knowledge. We argue that higher OS workers, even in high-risk occupations, are more likely to have access to (and to use) workplace COVID-19 risk mitigation measures than low OS workers because they are more likely: (a) to work for employers who voluntarily practice risk mitigation and provide PPE and other tools for workers to do so, (b) if necessary, to demand risk reduction measures and to have the bargaining power to obtain them, and (c) to understand (or learn about) COVID-19 transmission routes and to comply with risk reduction strategies or implement their own [26–28].

Once we disaggregate our results by occupational standing, we see large differences in the racial and ethnic distribution of frontline occupations. In contrast to White frontline workers, Latino, Black, and Native American frontline workers are overrepresented in lower OS

occupations overall, as well as in lower OS occupations associated with high risk, and thus are probably less likely than Whites to have adequate COVID-19 protections.

In contrast, Asians show more complex patterns, reflecting factors such as the heterogeneity of Asian groups, their immigration history, labor market segregation, and ethnic economic niches. For example, female Filipino workers are disproportionately employed in higher-risk occupations across all of the five risk categories. However, they are much more likely to be in higher OS occupations than other Asian groups, in part because of a history of immigration of Filipino nurses and other health professionals to the US [46].

Although most earlier studies found lower occupational risks for White workers compared with other groups, specific findings vary due largely to the types of occupations examined (e.g., frontline vs. essential) and the occupational characteristics chosen to denote potential risk for COVID-19. Based on data from the 2014–2017 Medical Expenditure Panel Survey and the 2017–2018 American Time Use Survey, Selden and Berdahl found that Black, Latino, and Asian workers were slightly more likely than Whites to work in essential jobs, i.e., at businesses that were allowed to remain open during COVID-19-related shutdowns, although essential White workers were more likely to be able to work from home [23]. Hawkins used data from the 2019 Current Population Survey linked with two measures of risk from O*NET (infection and close proximity) and found an elevated risk for Black workers but virtually none for Latinos for jobs in essential industries [9]. In a report by the Urban Institute based on the 2018 ACS that used O*NET data to identify occupations involving close proximity to other workers, researchers concluded that Black, Latino, and Native American workers were more likely than White workers to have jobs that put them at high risk for viral transmission [16]. Another study, based on employment data for 2014–2019 and only two O*NET measures (working indoors and in close proximity to others), found disproportionate risk exposure for both Black and Latino workers in some occupations [15].

In contrast to much of the earlier work, we focus on frontline occupations (i.e., those in which one-third or fewer workers can feasibly work from home) rather than essential industries. This choice is important because the definition of frontline occupations has remained relatively stable throughout the pandemic and across industries and jurisdictions, whereas occupations classified as essential have varied greatly among states and local jurisdictions and over time. We also expand measurement of potential risk of viral transmission used in previous work that has relied on only one or two occupation-related characteristics. However, our main contribution is that we distinguish among occupations by levels of occupational standing as a proxy for access to and use of COVID-19 risk reduction measures, a strategy that uncovered important differences among racial and ethnic groups that were not apparent in the previous studies.

The racial/ethnic differences by occupational standing highlighted in this paper are consistent with a large literature unrelated to the COVID-19 pandemic. Social science and public health research shows that the history and contemporary effects of racism in the US have led to a labor force highly segregated by race and ethnicity, with Latino and Black workers holding many of the lowest OS and least secure jobs [19, 20, 47, 48]. There is also ample evidence that workers in these occupations were already at higher risk of accidents, injury, infection, and other health problems prior to the pandemic and that these workers also have less control and decision-making ability over how their workplaces are run [49]. These workers may also be more likely to face employer resistance to implementing stricter safety measures that may reduce productivity or profits [47, 50–52]. Thus, a realistic depiction of workplace COVID-19 risks for Latino and Black workers requires consideration of occupational standing and/or careful direct measurement of implementation and regular use of COVID-19 mitigation strategies.

Despite the insights provided by this study, our analysis has several important limitations. First, the 2018 ACS data on recent jobs used here may not provide an accurate picture of employment or occupational distributions during the pandemic. Unemployment rates rose markedly in many areas of the country during the early part of 2020. For example, the rate was 14.7% in April 2020 compared to 3.5% in February 2020 for the US population age 16+ [53]. Unemployment rates were even higher for some racial/ethnic groups, including the Latino (18.9% in April) and Black populations (16.8% in May) [54, 55]. Business closures, layoffs, and unemployment rates have varied a great deal from state to state and over time. Thus, with currently available data it is impossible to determine whether or not workers were employed during 2020 at a particular time and place. Other consequences of the pandemic, such as virtual schooling, have led many parents, especially mothers, to leave the labor force entirely, at least until the pandemic is over [56]. Despite these caveats, we believe that the ACS data provide the best picture currently available of the occupational distribution of the US population at a time close to the onset of the pandemic. A related limitation is that, because our analyses provide no evidence about exposure for people who are unemployed or outside the labor force, our conclusions are necessarily restricted to the employed population.

As is true for all surveys, there is likely to be selective nonresponse in the ACS and misreporting of information (most relevant in our case are misreports of occupational classification and race/ethnicity). Moreover, the ACS occupational classification scheme consisted of a relatively modest number of occupational groups (409 total and 249 frontline), which required us to combine detailed O*NET occupations with potentially varying risk exposures into coarser occupational categories, thereby increasing the amount of intra-occupation variability.

There are also several limitations of the O*NET database and its specific measures we rely on to estimate workplace exposures to COVID-19. Responses to O*NET surveys may be affected by non-response and reporting errors, particularly given the subjectivity of many of the questions, but O*NET does not provide information on differential response rates. As an employer-based survey, O*NET measures the occupational characteristics of workers who are formally employed. Informal workers in our ACS sample, which likely include a substantial portion of immigrants and workers of color, probably have different working conditions than the formally employed workers who provided O*NET data. The omission of informal workers from O*NET may have led us to underestimate the extent of racial and ethnic differences in workplace exposures to COVID-19. In addition, O*NET measures were not designed with a pandemic in mind and thus do not include key information on viral transmission risk. A critical omission for analyses of COVID-19 is data on the implementation of, and worker compliance with, COVID-19 transmission workplace mitigation measures. This lack of information underscores the need to disaggregate workers by occupational standing or another proxy measure for mitigation, as we have done here. There are other limitations to the O*NET data when used in studies of infection-related risks. For example, in the case of contact with others, it would be useful to know the average duration of close contacts, the nature of the contacts and the characteristics of the people with whom the worker interacts. For all of our measures of risk, our analysis would benefit if O*NET provided information on how characteristics of each individual occupation vary by demographic factors (e.g., race/ethnicity, sex, and age of the workers).

Despite these limitations, our results strongly suggest that higher potential work exposures to COVID-19 likely contribute to a higher prevalence of the virus among Latino, Black, and Native American, compared to White, workers in the US. Even though vaccination rates have been increasing, risk reduction, particularly in settings, including workplaces, where transmission is high, must remain an important focus, especially given racial, ethnic, and social class

disparities in vaccination rates and the emergence of newer, more transmissible SAR-CoV2 variants.

Reducing potential exposure for workers in low OS occupations requires multiple coordinated measures beyond PPE. Carlsten and colleagues draw on industrial hygiene's hierarchy of controls framework to outline three types of measures employers should take to mitigate COVID-19 risks for their workers [17]. The first, eliminating exposure to the SARS CoV-2 virus completely, would clearly be the most effective measure–via options such as requiring vaccination, symptom and illness reporting, COVID-19 testing, providing paid sick leave that is ample and accessible, and providing telecommuting and work-from-home options. Second, engineering and administrative controls (e.g., physical barriers between people, improved ventilation systems, regular disinfecting protocols, and staggered work schedules) are the next most effective. Finally, provision and use of personal protective equipment (e.g., masks, shields) is the least effective, albeit essential (particularly combined with engineering and administrative controls) if eliminating exposure is not possible [17]. Analysis of survey data from June 2020 for non-health care workers who were unable to work remotely showed that voluntary use of COVID-19 workplace hazard controls (defined as PPE and other physical barriers) was approximately double when employers provided these hazard controls than when they did not [29]. The differences were particularly large among low-income workers, perhaps because they were less likely to be able to afford PPE themselves. All of these measures would additionally benefit the broader community through reduced transmission.

From a public policy point of view, the question is how to ensure that employers protect workers by implementing these risk mitigation strategies, particularly in workplaces employing lower OS workers. In the US, employers are legally required to provide a workplace free of recognized health and safety hazards. The federal Occupational Safety and Health Administration (OSHA), the Centers for Disease Control and Prevention (CDC) and state governmental agencies have provided guidance to employers about exposure controls against COVID-19 transmission [57], but enforcement has been limited [58]. Given the importance of workplaces as venues for transmission of COVID-19 and pathogens in future pandemics, federal and state agencies regulating worker safety need better tools and greater emergency power to act on behalf of worker health.

## Supporting information

**S1 Appendix. Racial/ethnic distributions within each occupational subset.**
(DOCX)

**S2 Appendix. Common high-risk occupations by occupational standing quartile.**
(DOCX)

## Acknowledgments

The authors would like to thank Francine Blau at Cornell University, Josefine Koebe at Universität Hamburg, and Pamela Meyerhofer at Montana State University for sharing their code to identify frontline occupations. We also thank Dana Glei and the anonymous reviewers for suggestions on earlier drafts of this paper.

## Author Contributions

**Conceptualization:** Noreen Goldman, Anne R. Pebley, Keunbok Lee, Theresa Andrasfay.

**Formal analysis:** Keunbok Lee, Boriana Pratt.

**Funding acquisition:** Noreen Goldman, Anne R. Pebley.

**Investigation:** Noreen Goldman, Anne R. Pebley, Keunbok Lee, Theresa Andrasfay.

**Methodology:** Noreen Goldman, Anne R. Pebley, Keunbok Lee, Theresa Andrasfay, Boriana Pratt.

**Project administration:** Noreen Goldman, Anne R. Pebley.

**Visualization:** Noreen Goldman, Anne R. Pebley, Keunbok Lee, Theresa Andrasfay, Boriana Pratt.

**Writing – original draft:** Noreen Goldman, Anne R. Pebley.

**Writing – review & editing:** Noreen Goldman, Anne R. Pebley, Keunbok Lee, Theresa Andrasfay, Boriana Pratt.

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
