## [Decision Letter · Decision Letter 0]

28 Jan 2021

PONE-D-20-36345

Racial and Ethnic DIfferentials in COVID-19-Related Job Exposures by Occupational Status in the US

PLOS ONE

Dear Dr. Goldman,

Thank you for submitting your manuscript to PLOS ONE. After careful consideration, we feel that it has merit but does not fully meet PLOS ONE’s publication criteria as it currently stands. Therefore, we invite you to submit a revised version of the manuscript that addresses the points raised during the review process.

Please submit your revised manuscript within 90 days of receipt of this email. If you will need more time than this to complete your revisions, please reply to this message or contact the journal office at plosone@plos.org. Please include the following items when submitting your revised manuscript:

We look forward to receiving your revised manuscript.

Kind regards,

Marlene Camacho-Rivera, ScD, MPH

Academic Editor

PLOS ONE

Journal Requirements:

Reviewers' comments:

Reviewer's Responses to Questions

**Comments to the Author**

1. Is the manuscript technically sound, and do the data support the conclusions?

Reviewer #1: Yes

Reviewer #2: Partly

2. Has the statistical analysis been performed appropriately and rigorously? 

Reviewer #1: N/A

Reviewer #2: Yes

3. Have the authors made all data underlying the findings in their manuscript fully available?

Reviewer #1: Yes

Reviewer #2: Yes

4. Is the manuscript presented in an intelligible fashion and written in standard English?

Reviewer #1: Yes

Reviewer #2: Yes

5. Review Comments to the Author

Reviewer #1: The manuscript is very well written. The use of “occupational status” is innovative and informative. The manuscript has many strengths and few minor weaknesses. The findings provide more evidence on the racial disparities experienced by workers of color during COVID-19 pandemic for policy considerations.

I just have a few suggestions:

- Typically, “occupational status” is defined by a combination of educational attainment and income. In this article, “occupational status” is defined by educational attainment only. Please comment on this point in your limitation section and whether your definition might or might not change your overall interpretation.

- Did you not have information on income?

- The text in the figures are blurry and hard to read.

References:

https://macses.ucsf.edu/research/socialenviron/occupation.php

Reviewer #2: Language tweak throughout: use Black workers instead of Blacks

Introduction: since the submission of this paper, there have been a few key studies about occupational risks by race/ethnicity and by industry. Suggest updating the lit review.

Introduction could be a lot meatier and get into some of the systemic drivers of potential reasons for greater exposure risk among low-wage and racial/ethnic minority workers. There’s been a ton of great work here—again, much of it published since this paper was submitted. But also the analysis needs more context and some data on, for example, disproportionate risks of hospitalizations and deaths by race and ethnicity, and hypothesized reasons. It also needs more citation in general, for statements like that in lines 85-87.

“Occupational status” is a very confusing term, and in reading the abstract I initially thought it was a typo. I assumed that it meant whether one was working or not working (synonymous with "employment status"), as it’s commonly used in that context in the occupational health literature. However, I see that you’re using it to mean “status” as in the social hierarchy, rather than status as in present/not present. I would suggest using a different term throughout.

In the intro, the authors appear to conflate race and ethnicity with “occupational status.” (lines 91-92). While the two often travel together, at this point in the manuscript they should be treated as separate unless they back up their assertions with the citations that appear later in the discussion.

Does the ACS include both occupation and industry for individual workers? When linking with o*net, do you consider industry, or just occupation? Industry could be quite important for establishing baseline level of risk by occupational setting. For example, a janitor in a hospital would be much more highly exposed than a janitor in a warehouse.

For “frontline status,” how do you account for the differential job loss that industries and occupations experienced during the pandemic? Yes, a server in a restaurant would be considered frontline because it can’t be done from home, but almost all servers lost their jobs—reducing their exposure status in the process. Not accounting for potential job loss could lead to substantial exposure misclassification

Table 1: suggest ordering by either size of the population or % frontline (for men) to make it easier to find the relevant information

The figures are nearly impossible to read, much less interpret. I printed out the manuscript and was unable to see the differences between the groups. In addition, readers are not good at comparing the relative size of bars on a figure aside from the categories at the beginning and the end, and it requires a heavy cognitive lift on the part of the reader. Strongly consider re-imagining how this data could be better visualized; I suggest any of Stephanie Evergreen’s books.

The point you make in lines 276-277 is precisely why industry, in addition to occupation, is crucial to consider. While educational attainment is also important, industry is available at an individual level and so there is lower risk of misclassification bias. From the discussion I understand the reasons that you chose to rely on occupation rather than industry (murky definitions of essential workers), but could you not use the combination of industry/occupation for the exposure assessment, and only industry for the comparison?

Re: the limitation of women being forced out of the labor market due to having young children, I imagine that the ACS measures both child age and marital status. How could the authors leverage this information to do a sensitivity analysis?

In the paragraph beginning on line 433, there’s a highly cited framework called the NIOSH Hierarchy of Controls that refers to different strategies for risk reduction/mitigation. Suggest citing it here and using it as an organizing framework for this paragraph/section.

Finally, any analysis that looks at racial disparities should explicitly discuss systemic racism as an upstream driver of the distribution of race/ethnicity by occupations.

Overall, this manuscript is very much in need of a coauthor or collaborator with expertise in occupational epidemiology, especially when using a job exposure matrix such as onet. There are clear gaps in knowledge (use of o*net, hierarchy of controls, intersection of occupation and industry classifications, testing for precision and accuracy of the assessed exposure) that should be addressed in a paper that explicitly addresses occupational exposure burden. I also suggest adding a social epidemiology coauthor or collaborator with experience in applying health disparities frameworks.

6. PLOS authors have the option to publish the peer review history of their article (what does this mean?). If published, this will include your full peer review and any attached files.

Reviewer #1: No

Reviewer #2: No

---

## [Author Response · Author response to Decision Letter 0]

6 Apr 2021

We thank both reviewers for their helpful comments that we believe strengthened our paper. Below we present each of the comments (in italics) followed by our response.

Reviewer #1 

• The manuscript is very well written. The use of “occupational status” is innovative and informative. The manuscript has many strengths and few minor weaknesses. The findings provide more evidence on the racial disparities experienced by workers of color during COVID-19 pandemic for policy considerations.

We thank the reviewer for this positive feedback. 

• I just have a few suggestions:

Typically, “occupational status” is defined by a combination of educational attainment and income. In this article, “occupational status” is defined by educational attainment only. Please comment on this point in your limitation section and whether your definition might or might not change your overall interpretation. Did you not have information on income?

Duncan’s socioeconomic status index (SEI) does indeed combine occupational education and occupational status into a weighted index. However, Hauser and Warren (1997) – cited in the paper and below – argued persuasively that occupational education and occupational income should be used as two separate measures because they have different effects and much of the literature now does so. Our goal was to identify a simple proxy for the amount of power and control workers are likely to have in determining how they do their work and whether they have access to PPE and other workplace protections. We do have information on income, but we chose occupational education alone for this purpose to avoid complicating our results further by using two SES separate indices. Occupational education is easy to understand, frequently used in the sociological literature, and, compared to occupational income, less likely to have been altered by the pandemic. Furthermore, education is typically completed at young ages, whereas job choices (often based heavily on wage and salary rates) are made throughout the working years – so occupational income is more endogenous to the choice of job than is occupational education. 

We have now changed the term we use from “occupational status” to “occupational standing” to avoid confusion.

We have revised our discussion of this issue in the text. 

Reference: Hauser RM, Warren JR. (1997) Socioeconomic Indexes for Occupations: A Review, Update, and Critique: Sociological Methodology. Available: https://journals.sagepub.com/doi/10.1111/1467-9531.271028

• The text in the figures are blurry and hard to read.

We have completely revamped the figures. The figures now directly examine the overrepresentation vs. underrepresentation of frontline workers in high-risk occupations and show the results both numerically and by color (using a heatmap). These graphs are much easier to read and to interpret than the previous figures. 

Reviewer #2

• Language tweak throughout: use Black workers instead of Blacks

Thanks for catching this. We have fixed this throughout the paper.

• Introduction: since the submission of this paper, there have been a few key studies about occupational risks by race/ethnicity and by industry. Suggest updating the lit review.

We have updated the literature review (primarily in the Introduction and Discussion sections), and now include studies published through March 2021 .

• Introduction could be a lot meatier and get into some of the systemic drivers of potential reasons for greater exposure risk among low-wage and racial/ethnic minority workers. There’s been a ton of great work here—again, much of it published since this paper was submitted. But also the analysis needs more context and some data on, for example, disproportionate risks of hospitalizations and deaths by race and ethnicity, and hypothesized reasons. 

It also needs more citation in general, for statements like that in lines 85-87.

As noted above, we have revised the introduction to include new papers. We have also added language that refers to the “the long history and contemporary effects of structural racism on occupational segregation in the US” and provided references. 

We agree that we could have written a much broader introduction discussing the role of structural racism and other social processes in producing the racial and ethnic distribution of workers by occupation, the reasons that lower wage workers are less likely to have access to PPE, and the disproportionate risks of hospitalizations and deaths. These are issues that we have written about at length in other work. However, this paper is specifically focused on the empirical evidence about the potential exposure to COVID-19 of frontline workers in different racial and ethnic groups. There has been extensive discussion in the press and academic publications about the issue, but limited empirical evidence – particularly evidence that includes proxy information on the likelihood of risk reduction mitigation, as we try to provide in this paper. We believe that it is essential to make these key empirical findings available to as broad an audience as possible as quickly as possible and, therefore, we have kept the narrow focus. 

• “Occupational status” is a very confusing term, and in reading the abstract I initially thought it was a typo. I assumed that it meant whether one was working or not working (synonymous with "employment status"), as it’s commonly used in that context in the occupational health literature. However, I see that you’re using it to mean “status” as in the social hierarchy, rather than status as in present/not present. I would suggest using a different term throughout.

At your suggestion, we have changed the term to “occupational standing” throughout the paper, figures and appendices. “Occupational status” is a term from the sociological and social science literature used to refer to an individual’s position in the occupational hierarchy. However, we agree that it can be a confusing term for readers in other fields and so we have made the change. 

• In the intro, the authors appear to conflate race and ethnicity with “occupational status.” (lines 91-92). While the two often travel together, at this point in the manuscript they should be treated as separate unless they back up their assertions with the citations that appear later in the discussion.

The lines referred to in the reviewer’s comment read: 

91 Examining results by occupational status leads to a clearer picture of racial/ethnic differentials in

92 potential exposure to COVID-19 transmission.

Note that “occupational status” originally used in line 91 is now “occupational standing.” Our point was that differences in the distribution of occupations by race and ethnicity are likely to contribute to differences in exposure to COVID-19, since Black, Latino, and some other groups of workers are more likely to work in jobs without adequate protection compared to whites and some other groups. We are not in any way conflating race and ethnicity with occupational (or socioeconomic) status. The sentence in lines 91-92 was proceeded by the following text, which we believe makes the meaning of the text in 91-92 clear:

“In contrast, higher status workers have better access to risk mitigation measures such as personal protective equipment (PPE), frequent sanitation, enforced distancing, partitions, better ventilation, and new filtration systems.”

• Does the ACS include both occupation and industry for individual workers? When linking with o*net, do you consider industry, or just occupation? Industry could be quite important for establishing baseline level of risk by occupational setting. For example, a janitor in a hospital would be much more highly exposed than a janitor in a warehouse.

We agree that use of PPE and other forms of protection are likely to vary by industry as well as occupation. The ACS data contains both variables. However, the O*NET data provides exposures only for occupations (see https://www.onetcenter.org/overview.html). Thus, even if we subdivide ACS respondents both by occupation and industry, the exposure information we obtain from O*NET would be identical for respondents in the same occupation, regardless of industry.

• For “frontline status,” how do you account for the differential job loss that industries and occupations experienced during the pandemic? Yes, a server in a restaurant would be considered frontline because it can’t be done from home, but almost all servers lost their jobs—reducing their exposure status in the process. Not accounting for potential job loss could lead to substantial exposure misclassification

We agree that this is a potential problem, in our analysis and in all those which have sought to use pre-pandemic data to examine potential effects of the pandemic. The issue is even more complicated because “job loss” for individual workers changed a great deal during the pandemic depending on what was and was not open. In many states, some businesses were closed, then reopened for a while, then closed again, then reopened, etc. Many of these business that were unable to keep workers on the payroll while closed (although the Payroll Protection Program initially helped a lot of employers continue to pay workers) reemployed them when they reopened. The detailed data to take this issue into account is not yet available. However, we added a sentence to the text that refers to differences by race and ethnicity in job losses during the pandemic, and we specifically acknowledge our inability to account for job losses and unemployment. 

With regard to the broader point about exposure misclassification, throughout the text we are now careful not to imply that these measures correspond to an individual’s experienced risk during the pandemic; rather, they correspond to racial and ethnic patterns of employment in high-risk occupations. 

• Table 1: suggest ordering by either size of the population or % frontline (for men) to make it easier to find the relevant information

We believe that it is important to keep the order of racial/ethnic groups consistent with the graphs (and appendix tables) as well as to have the groups that have received the most attention listed first (White, Black and Latino) and to keep the four Asian groups together. We have changed the placement of Native Americans to come after the four Asian groups and directly before Asian and Pacific Islanders. This ordering is now consistent in Table 1, the figures, and the appendices.

• The figures are nearly impossible to read, much less interpret. I printed out the manuscript and was unable to see the differences between the groups. In addition, readers are not good at comparing the relative size of bars on a figure aside from the categories at the beginning and the end, and it requires a heavy cognitive lift on the part of the reader. Strongly consider re-imagining how this data could be better visualized; I suggest any of Stephanie Evergreen’s books.

As we noted in our response to Reviewer 1, we have completely redone the figures. The reader no longer is asked to assess the relative size of bars (an admittedly difficult and unreasonable task); these relative measures are computed directly and are shown as measures of overrepresentation and underrepresentation in various categories of occupations. Interpretation is aided by the use of a heatmap and a detailed numerical example.

• The point you make in lines 276-277 is precisely why industry, in addition to occupation, is crucial to consider. While educational attainment is also important, industry is available at an individual level and so there is lower risk of misclassification bias. From the discussion I understand the reasons that you chose to rely on occupation rather than industry (murky definitions of essential workers), but could you not use the combination of industry/occupation for the exposure assessment, and only industry for the comparison?

The lines referenced (276-277) have been rewritten to better clarify our focus on occupational standing and avoid comparing across industry. We now provide examples of two occupations within the healthcare sector—home health aides and physicians—to emphasize how occupations with similar levels of risk factors according to O*NET can have very different statuses. We have also included some citations to recent work to support this claim that lower status healthcare workers have received inadequate training in infectious disease protocol and faced severe shortages of personal protective equipment.

As noted above, only occupation-related exposure information is available from O*NET so we are unable to directly consider how these risk factors vary across industry. 

• Re: the limitation of women being forced out of the labor market due to having young children, I imagine that the ACS measures both child age and marital status. How could the authors leverage this information to do a sensitivity analysis?

An analysis of women who had to leave the labor market to take care of children when schools were closed is very important, but is beyond the scope of this paper. From media and emerging research, we know that parents have made a huge variety of arrangements for children during this period, which have been affected by the type of child care (or after school care) arrangements they had pre-pandemic, the availability of family close by, services offered by community organizations, etc. With our available data, we do not have the ability to consider all the relevant factors necessary to make predictions about which women were likely to exit the labor force.

• In the paragraph beginning on line 433, there’s a highly cited framework called the NIOSH Hierarchy of Controls that refers to different strategies for risk reduction/mitigation. Suggest citing it here and using it as an organizing framework for this paragraph/section.

This part of the discussion section is now rewritten and refers both to the HoC and to a thoughtful article applying this framework to COVID-19 hazard mitigation.

• Finally, any analysis that looks at racial disparities should explicitly discuss systemic racism as an upstream driver of the distribution of race/ethnicity by occupations.

Concerns about systemic racism and its consequences for workers’ location in the employment hierarchy and for health led us to undertake this research. Our research team has a long track record of research on racism, discrimination, and social inequality and health outcomes. However, in this paper, in order to complete and publish the research while it still can be useful to readers, we decided to focus on the effects of the occupational distribution of workers by race and ethnicity on exposure rather than on the causes of the occupational distribution of workers by race and ethnicity itself. 

While we note the role of racism in occupational segregation, providing an adequate treatment of the social processes, including systemic racism, which have led to current occupational disparities by race and ethnicity would require a considerable lengthening of the paper and additional empirical analyses that would delay the availability of these results and greatly enlarge the scope of the paper.

• Overall, this manuscript is very much in need of a coauthor or collaborator with expertise in occupational epidemiology, especially when using a job exposure matrix such as onet. There are clear gaps in knowledge (use of o*net, hierarchy of controls, intersection of occupation and industry classifications, testing for precision and accuracy of the assessed exposure) that should be addressed in a paper that explicitly addresses occupational exposure burden. I also suggest adding a social epidemiology coauthor or collaborator with experience in applying health disparities frameworks.

The authors of this paper comprise a multidisciplinary research team whose members include epidemiologists and social scientists with an extensive track record of research on health disparities by race, ethnicity, and immigration status and the social causes of occupational attainment and social position. We also have considerable experience with analyses of O*NET, occupational and industry classifications, and occupational exposures. As noted above, this paper is intentionally sharply focused because, given our experience and background, we believe that it is essential to provide solid empirical results on the extreme disadvantage in terms of exposure to COVID-19 faced by many frontline workers during the pandemic, particularly among Black, Latino, Native American, and some other groups of workers.

---

## [Decision Letter · Decision Letter 1]

21 Jun 2021

PONE-D-20-36345R1

Racial and Ethnic DIfferentials in COVID-19-Related Job Exposures by Occupational Standing in the US

PLOS ONE

Dear Dr. Goldman,

Thank you for submitting your manuscript to PLOS ONE. After careful consideration, we feel that it has merit but does not fully meet PLOS ONE’s publication criteria as it currently stands. Therefore, we invite you to submit a revised version of the manuscript that addresses the points raised during the review process.

Please submit your revised manuscript within 90 days of receipt of this email. If you will need more time than this to complete your revisions, please reply to this message or contact the journal office at plosone@plos.org. Please include the following items when submitting your revised manuscript:

We look forward to receiving your revised manuscript.

Kind regards,

Marlene Camacho-Rivera, ScD, MPH

Academic Editor

PLOS ONE

Reviewers' comments:

Reviewer's Responses to Questions

**Comments to the Author**

1. If the authors have adequately addressed your comments raised in a previous round of review and you feel that this manuscript is now acceptable for publication, you may indicate that here to bypass the “Comments to the Author” section, enter your conflict of interest statement in the “Confidential to Editor” section, and submit your "Accept" recommendation.

Reviewer #1: All comments have been addressed

Reviewer #3: (No Response)

Reviewer #4: (No Response)

2. Is the manuscript technically sound, and do the data support the conclusions?

Reviewer #1: Yes

Reviewer #3: Partly

Reviewer #4: Partly

3. Has the statistical analysis been performed appropriately and rigorously? 

Reviewer #1: Yes

Reviewer #3: I Don't Know

Reviewer #4: Yes

4. Have the authors made all data underlying the findings in their manuscript fully available?

Reviewer #1: Yes

Reviewer #3: Yes

Reviewer #4: Yes

5. Is the manuscript presented in an intelligible fashion and written in standard English?

Reviewer #1: Yes

Reviewer #3: Yes

Reviewer #4: Yes

6. Review Comments to the Author

Reviewer #1: The revision adequately addresses all the comments. I appreciate reading the nuances of the data. Well done.

Reviewer #3: PONE-D-20-36345R1

Racial and Ethnic Differentials in COVID-19-Related Job Exposures by Occupational Standing in the US

Thank you for the opportunity to review this revised manuscript. The authors present an interesting descriptive exercise trying to disentangle the occupational and ethnic/racial heterogeneity in exposure and risk of COVID-19. The document is well written and describes clearly the analytical process followed to arrive to their results.

Although the findings are very interesting and clearly presented, I consider that there are few methodological aspects that are key for the observed results and the interpretation, that are missing in this document. This is, confounding, the presence of heterogeneity and intersectionality. Given that the authors did not conduct any confounding adjustment, the robustness of the estimates is questionable. The authors did, however, stratify the analyses by the occupational standing and race/ethnicity, which highlights the issues with heterogeneity and the intersectionality (related to systemic racism, discriminations, opportunities for education, access to health care, and access and correct implementation of PPE, etc.) and call the attention to think about other several factors that could explain the results and that are not accounted for in this study. Therefore, I consider that as an initial approach to shed light into the issue of occupation and ethnicity in the disproportionate presence of COVID-19 outcomes among Black and Latinos, this is an adequate exercise. However, to properly address this issue, I would recommend a more robust approach, which would likely require other type of analysis and perhaps other type of data, which is beyond the scope of this manuscript. Hence, although I favour sharing this document with the academic community -again as a first step to identify the intricacies in the intersection between occupation and race/ethnicity in COVID-19- I would recommend some revisions.

Abstract:

In the body of the main manuscript the authors indicate:

Page 13; lines 268- 273 “Contrary to expectation based on earlier studies and the

media, the values in Fig 1 suggest that Black and Latino workers do not face higher occupational risks related to viral exposure compared with Whites. However, this conclusion is likely to be incorrect. The classification of high-risk occupations that we have considered thus far does not capture variation among occupations in COVID-19 risk reduction measures that are deployed in the workplace and are more likely to be implemented in higher status occupations.”

This aspect is key, the cumulative results (i.e, from the non-stratified analysis) are biased! Since the abstract is the door to the main manuscript, I recommend incorporating a sentence indicating that the overrepresentation of White in high-risk occupations is artifactual and incorrect, as you did in the results section of the manuscript. Likewise, I would recommend the consistent use of “lower (or higher) occupational standing” instead of “lower (or higher) occupational status”, to be consistent with the terminology used throughout the main text.

Introduction:

Line 74, please provide a reference for the statement and provide a short sentence about what has been considered essential vs frontline (you described “frontline” in page 8, but the “essential” definition is not provided).

Data:

Line 112: the ethics statement could be placed at the beginning or at the end of the section. In the current state, it cuts the flow for the information on the data sources.

Line 121: what are the six variables? I read the five types of “risk” but not the variables.

Race/ethnicity:

Do the authors consider that the “other Asian” category could be homogeneous? For instance, are Japanese and Pakistani similar groups? What could be the impact of merging these groups? How could this explain the results from Figure 4? Similar thoughts for “other race/ethnicity”. This could be mentioned in the discussion.

Line 155, please provide references for this statement.

Defining high risk occupations:

Lines 178-180. Which responses are you referring to? Also, could you indicate the implication of the mentioned correlation in the responses?

Analytic strategy:

Line 198-199: What could be the direction of the bias in presence of a differential response rate by ethnicity in the O*NET coupled with the differential labor participation rates? would suggest indicating this in the limitations.

I would recommend providing details of the analysis, including exclusion of observations, presence of missing data if any, software used for calculations and to generate the figures, etc.

Discussion

Line 371-373: using occupational standing for all the mentioned workers and workplace exposure to COVID-19 is an overstatement. There are several other aspects related to access, use and implementation of measures to decrease exposure to COVID-19 that are not solely based on occupational standing.

Limitations: I would recommend including the above-mentioned issues related to lack of adjustment for confounding, absence of evaluation of heterogeneity or interaction across variables and intersectionality, and the potential effects in the results and their interpretation. Likewise, please mention issues related to the misclassification of the work and therefore occupational standing?

Line 481-492: maybe goes beyond the scope of the manuscript and could be summarized.

Reviewer #4: Introduction

-The key “work-related factor” you are missing here is pay and access to healthcare. This means less ability to get care, less ability to take paid time off (increasing presenteeism), higher likelihood of living in crowded housing etc. Also for your point 1: Black, Latino and NA more likely to hold jobs that have to be done at their workplace rather than remotely…I am not sure this is totally true, it’s jobs that can’t be done from home which could mean they are working during the pandemic or could mean they have lost their job altogether (See Baker 2020 in AJPH: https://ajph.aphapublications.org/doi/full/10.2105/AJPH.2020.305738)

-Line 66: “we investigate these claims”—previously you cited peer-reviewed literature, seems reasonable to refer to that as more than a claim. Maybe “we investigate these hypotheses” or “these findings”. Claims makes it seems like you are telling occupational health researchers they are just making up things.

-Is there a reason you aren’t including median pay in your analysis? That is very easy to access and would give more weight to the “occupational standing”—and it’s easy to integrate with O*NET data

-Line 93: “lower status workers”—is this really want you mean? It’s only higher status because we as educated people have decided what is higher v. lower status work. Status really more an opinion than something that can be measured. Maybe “lower paid”, “more vulnerable”, “precarious” “underrepresented” ??

-Line 95: yes, underrepresented workers in more precarious jobs often have less access to risk mitigation strategies and workplace POWER than those in higher paid job. Worth mentioning the power element here. Maybe no one is getting proper PPE but there are some workers who could more easily ask than others

Data

-Can you defend why you used 1 year ACS data instead of 5 year? Especially for small demographic groups, 1 year is extremely noisy—5 year would be better. When you start stratifying by groups you start to lose reliability of the estimate with only 1 year ACS data.

See: https://www.census.gov/content/dam/Census/library/publications/2018/acs/acs_general_handbook_2018_ch03.pdf

-Worth noting O*NET isn’t just a sample of workers—also includes employers and job experts. (line 114)

-Line 120: virus is primarily transmitted through aerosols. This should at least be listed first (before respiratory droplets).

Results

-I’m quite confused about the numbers in your sample set. There are 160+ million workers in the USA. How come you only have so few? Given many of the groups that you report (Chinese, Filipino, Vietnamese, etc.) have less than 65,000 people, the 1 year ACS data is likely not very valuable and you should be using 5 year data instead to have more valid estimates.

-Line 251-253: this type of commentary seems more like discussion section fodder

-Your results assume that the (rudimentary) exposure metrics you included capture all occupational exposure risk with statements like “…Black and Latino workers do not face higher occupational risks related to viral exposure compared with Whites.” Instead what you mean to say is …”Black and Latino workers are less likely to be employed in occupations with characteristics related to increased exposure to SARS-CoV-2, as ascertained from five O*NET metrics.” Your clarifying sentence after this is also helpful and welcomed! However, do I think more of this should be in the discussion as opposed to in the results.

-Line 279: And be paid a living wage, have access to a union, etc.

Discussion

-I struggle (as another reviewer did) with saying that years of college education is a reasonable proxy for “occupational standing”. I think at a minimum wage should also be considered. This to me is a very “white collar” interpretation of occupational standing. But in the trades, where workers enjoy high pay, strong union protections, and access to benefits/safety…most have not completed a year of college. Pay/union representation seem equally important to me. Instead of using just the one variable as a proxy for “occupational standing” instead just refer to it as “educational attainment” with an understanding that we are imposing very white collar values on what makes “good work” and “important work”

-Line 375: “higher status”—suggest eliminating this phrasing as detailed in a previous comment

-Line 377: this gets at some of what I was detailing earlier, but I don’t think this necessarily relates to years of college. The trades/all goods producing occupations would fall into these categories even if their workers didn’t go to college (unless trade school is counted as college, but that seems unlikely)

-Line 383—another stray “status”

-Line 409: I get that there are differences between “essential” and “frontline” (very militaristic way of referring to these workers!) but wouldn’t the same limitations apply? “Frontline” would also have variability by states and over time? For example, if a restaurant is closed then those are not “frontline” workers, but if open then they would be, wouldn’t they?

Limitations

-Limitations with using 1 year ACS v. 5 year ACS especially for the small groups you are considering?

-Limitations because using the ACS means you have to combine some detailed occupations so you miss out on variability between occupations?

-Only able to look at a small subset not the whole workforce

-O*NET limitations: doesn’t cover all occupations

-O*NET limitations: subject to misclassification and undercounting, generated from subjective questionnaires, differential misclassification across groups (e.g. folks may not know they are exposed to disease/infection unless they are told they are and those risks are communicated making people in healthcare more likely to realize than other occupations), within-job exposure variation is not accounted for

-General limitations of your study: Many, many occupations/people are missing (some of which would be considered “frontline” and/or lower standing): independent contractors, domestic workers, military, is agriculture included?, self-employed, contingent, undocumented…These are also jobs that are more diverse

-General limitation: don’t include wages which is likely a larger predictor for how a pandemic would affect a worker

-Line 468: Disagree. If employers reduce workplace transmission, then community transmission is also reduced. We have to stop separating “occupational health” and “public health”. They are interconnected and if we protect workers, we are positively benefitting the community.

-Line 470: Thank you for giving some examples! However, these examples really only work for well paid, likely white workers. “preventing workers with symptoms or positive tests from coming to work” Does little for the Black taxi driver who has to make money in order to feed his family and taking a week off work could financially devastate him. How about instead, “providing paid sick leave that is ample and accessible, for which workers do not face retaliation or retribution for taking, if they have an exposure event” Or something like that.

-Line 475: is this still true about testing? I think we are at a point where we have a range of rapid and accurate tests…and isn’t all testing free even to the uninsured? People might not realize that, but it is.

-Line 486: Exposure controls are all the controls you are mentioning? Delete exposure. Also, “employers can provide PPE?” How about employers must or employers should

Figure 1

-Very blurry. Would be helpful to have the x-axis on the male section as well, it can be hard to follow it all the way down to the bottom

7. PLOS authors have the option to publish the peer review history of their article (what does this mean?). If published, this will include your full peer review and any attached files.

Reviewer #1: No

Reviewer #3: No

Reviewer #4: No

---

## [Author Response · Author response to Decision Letter 1]

8 Jul 2021

We thank the reviewers for their helpful comments. Below we provide point-by-point responses to each of the reviewer’s comments. The reviewer’s comments are presented in italics (in the Word file).

Reviewer 1:

No further comments or corrections

Reviewer 3:

Although the findings are very interesting and clearly presented, I consider that there are few methodological aspects that are key for the observed results and the interpretation, that are missing in this document. This is, confounding, the presence of heterogeneity and intersectionality. Given that the authors did not conduct any confounding adjustment, the robustness of the estimates is questionable. The authors did, however, stratify the analyses by the occupational standing and race/ethnicity, which highlights the issues with heterogeneity and the intersectionality (related to systemic racism, discriminations, opportunities for education, access to health care, and access and correct implementation of PPE, etc.) and call the attention to think about other several factors that could explain the results and that are not accounted for in this study. Therefore, I consider that as an initial approach to shed light into the issue of occupation and ethnicity in the disproportionate presence of COVID-19 outcomes among Black and Latinos, this is an adequate exercise.

The objective of this analysis is to extend previous research which attempts to quantify work-related exposure to COVID-19 by race and ethnicity. Neither these previous studies nor our study attempts to provide a causal analysis to explain differences in risk across racial/ethnic groups by demographic and socioeconomic characteristics of the groups (e.g., by estimating multiple regression models) and so confounding is not an issue here. The goal of stratification by occupational standing is to provide a clearer picture of racial/ethnic differences in potential exposure to COVID-19 transmission. Occupational standing serves as a proxy (albeit an imperfect one) for workplace risk mitigation measures. We recognize that there are many variables that are important determinants of differential risk across racial and ethnic groups, such as those mentioned by the reviewer and those derived from factors beyond occupational exposure, but this type of analysis is not consistent with the objectives of this paper.

Page 13; lines 268- 273 “Contrary to expectation based on earlier studies and the

media, the values in Fig 1 suggest that Black and Latino workers do not face higher occupational risks related to viral exposure compared with Whites. However, this conclusion is likely to be incorrect. The classification of high-risk occupations that we have considered thus far does not capture variation among occupations in COVID-19 risk reduction measures that are deployed in the workplace and are more likely to be implemented in higher status occupations.”

This aspect is key, the cumulative results (i.e, from the non-stratified analysis) are biased! Since the abstract is the door to the main manuscript, I recommend incorporating a sentence indicating that the overrepresentation of White in high-risk occupations is artifactual and incorrect, as you did in the results section of the manuscript. Likewise, I would recommend the consistent use of “lower (or higher) occupational standing” instead of “lower (or higher) occupational status”, to be consistent with the terminology used throughout the main text.

This issue is related to the previous concern. The non-stratified results are not biased nor is the overrepresentation of Whites in high-risk occupations artifactual or incorrect. These results portray the riskiness of occupations by race and ethnicity in the US population. A central point of our paper is precisely that this simple picture can be misleading because it ignores the fact that some race/ethnic groups in potentially high-risk jobs (e.g., physicians) are more likely to be protected at work than others. For example, Filipino frontline workers have a greater overrepresentation in occupations involving high risk of infection (such as healthcare work) than all other racial and ethnic groups. But, when we control for occupational standing, it becomes clear than many Filipinos in high risk environments are doctors and nurses. Our point is that although workers in higher occupational standing jobs may be employed in potentially hazardous environments (e.g., around many people with COVID-19), they are also much more likely than other workers to have protection, e.g., PPE, time and places to wash, highly frequent cleaning, etc., that reduces their risk of infection. Whether protection against COVID-19 transmission is available to workers is not captured in O*NET or other national data on workplace conditions. For this reason, we use occupational standing as a proxy for protection within the workplace. However, none of this negates the overall large overrepresentation of Filipinos in occupations with a high risk of infection, as derived from the O*NET measures.

We agree with the reviewer that the use of the term “occupational status” can be problematic. We have changed occupational status to occupational standing (OS) and, to avoid cumbersome text, we now use the terms low OS occupations and high OS occupations throughout the text in lieu of low status and high status occupations. It is important to keep in mind that the designation high or low standing does not refer to individual people, but rather to the standing of the occupation they hold compared to other occupations as based on the educational attainment of a national sample of workers in each occupation.

Introduction:

Line 74, please provide a reference for the statement and provide a short sentence about what has been considered essential vs frontline (you described “frontline” in page 8, but the “essential” definition is not provided).

We have added a sentence with several citations that provide different definitions of essential occupations. There has been greater (although not complete) agreement on the definition of frontline than of essential. 

Data:

Line 112: the ethics statement could be placed at the beginning or at the end of the section. In the current state, it cuts the flow for the information on the data sources.

We moved the ethics statement to the end of the section.

Line 121: what are the six variables? I read the five types of “risk” but not the variables.

We agree that this was confusing as written. We have revised several sentences in the Data section to clarify this. Two variables are used to capture working indoors. Risk 5 in the subsection “Defining high risk occupations” presents the two questions.

Race/ethnicity:

Do the authors consider that the “other Asian” category could be homogeneous? For instance, are Japanese and Pakistani similar groups? What could be the impact of merging these groups? How could this explain the results from Figure 4? Similar thoughts for “other race/ethnicity”. This could be mentioned in the discussion.

We definitely do not consider the “Other Asian” group to be homogenous. The ACS does not provide information regarding the groups that comprise the “Other race/ethnicity” category, but it is undoubtedly also a heterogeneous group. We have added “heterogenous” to the appropriate sentence in the text. Because of this heterogeneity and the relatively small size of these groups, we do not focus on them in the analysis. 

Line 155, please provide references for this statement.

We added several references on this issue; we have placed them in the previous paragraph.

Defining high risk occupations:

Lines 178-180. Which responses are you referring to? Also, could you indicate the implication of the mentioned correlation in the responses?

We are referring to responses to the two questions on working outdoors (shown under risk 5). We have clarified this in the text. The strong correlation is a good sign since it suggests that both of these questions are capturing the same types of occupations.

Analytic strategy:

Line 198-199: What could be the direction of the bias in presence of a differential response rate by ethnicity in the O*NET coupled with the differential labor participation rates? would suggest indicating this in the limitations.

Unfortunately, O*NET does not provide information on the demographic characteristics of its survey respondents, so we have no idea whether non-response varies by race and ethnicity (or any other characteristics). The overall O*NET response rate is around 63% (O*NET, 2021) so it is possible that some groups were less likely to respond than others. Figuring out the consequences of potential differential underreporting in the absence of such information is complex. For example, if Latino workers were less likely to complete the O*NET survey and also more likely to be exposed to the general public or to work in close contact with coworkers than others in their same occupation, the potential risk score for that occupation would be artificially low. In contrast, if occupational incumbents included very few Latinos, the bias from Latino non-response would be minimal. Potential differential labor force participation rates add further complexity. While the reviewer raises a good point, it is impossible to provide meaningful estimates of the direction or magnitude of these types of biases. However, we have acknowledged the potential for non-response (from both the ACS and O*NET) in the text.

O*NET. 2021. Appendix E: Summary of Response Rate Experience to Date. https://www.onetcenter.org/dl_files/omb2021/AppendixE.pdf

I would recommend providing details of the analysis, including exclusion of observations, presence of missing data if any, software used for calculations and to generate the figures, etc.

We added information in the Analytic Strategy section on the number of persons in the ACS with military occupations (i.e., those excluded from the analysis because of the absence of O*NET information). No individuals in our sample of employed workers were missing information on education or race/ethnicity in the files provided by the US Census Bureau. We also added a sentence at the end of the Analytic Strategy indicating that STATA/MP Version 15.1 was used for calculations and figure production.

Discussion

Line 371-373: using occupational standing for all the mentioned workers and workplace exposure to COVID-19 is an overstatement. There are several other aspects related to access, use and implementation of measures to decrease exposure to COVID-19 that are not solely based on occupational standing.

As we say in the paper, we use occupational standing as a proxy for COVID-19 workplace mitigation efforts, not to represent occupational exposure nor as a determinant of workplace protections. As we discuss later in this section, we recognize that this is an imperfect measure that is subject to numerous limitations that we describe. Ideally, O*NET or another data source would have collected information on the implementation and regular use of workplace protections directly. In the absence of such data, we argue that occupational standing is a reasonable proxy for worker access to COVID-19 mitigation measures because extensive public health literature shows that “those at higher socioeconomic status… generally have access to a wider array of resources, including power and influence, to protect health than others do.” (pg. 5 in the manuscript). Specifically, we argue that “…higher [occupational standing] workers are more likely: (a) to work for employers who voluntarily practice risk mitigation and provide PPE and other tools for workers to do so, (b) if necessary, to demand risk reduction measures and to have the bargaining power to obtain them, and (c) to understand (or learn about) COVID-19 transmission routes and to comply with risk reduction strategies or implement their own.” In the Discussion section, we elaborate on the types of information that would have been desirable but are simply not available in O*NET – including the important factors that you mention.

Limitations: I would recommend including the above-mentioned issues related to lack of adjustment for confounding, absence of evaluation of heterogeneity or interaction across variables and intersectionality, and the potential effects in the results and their interpretation. Likewise, please mention issues related to the misclassification of the work and therefore occupational standing?

We agree that adjustment for confounding, evaluation of heterogeneity and consideration of interactions would be important if this paper were intended as a causal or explanatory analysis. As we now indicate more explicitly at the beginning of the paper: “Our goal is to contribute insights into the pandemic by presenting the size and scope of these racial and ethnic disparities among workers rather than by estimating causal models of their determinants.” Because this is a descriptive analysis, confounding is not a problem, nor is the lack of interactions. In terms of intersectionality, the key innovation of our analysis (introducing OS as a proxy for risk mitigation) is a form of intersectional analysis: previous research did not provide as complete a picture as we do because we look at the intersection of race and ethnicity with OS (as well as gender). By considering the intersection of these variables, we develop a clearer picture of race and ethnic differentials in potential COVID-19 risk.

On the misclassification of work: respondents are not asked to pick an occupation from a list presented to them – many years of research have shown that this would lead to poor occupational reporting. Instead, respondents provide detailed verbal descriptions of their jobs and answer questions about their responsibilities and activities, the type of organization they work for or if they are self-employed, their job title, and the industry they work in. Trained occupational coders at the US Census Bureau use the verbal descriptions and other responses and a standardized system for occupational coding to code each respondent’s occupation. (The ACS questions (41 to 46) used to collect information for occupation coding are on page 11 at https://www2.census.gov/programs-surveys/acs/methodology/questionnaires/2018/quest18.pdf). Of course, even with these procedures, occupations can be misclassified, in the same way other data in surveys can be misreported by respondents. However, the use of detailed verbal work descriptions and standardized classification procedures using these verbal descriptions by trained classifiers reduces this potential. We now acknowledge in the Discussion section that misclassification of occupation could occur.

Line 481-492: maybe goes beyond the scope of the manuscript and could be summarized.

We added this section to the discussion at the explicit request of a PLOS ONE reviewer for the last round of reviews of this paper. We agree that it can be summarized and have done so.

Reviewer #4: 

Introduction

-The key “work-related factor” you are missing here is pay and access to healthcare. This means less ability to get care, less ability to take paid time off (increasing presenteeism), higher likelihood of living in crowded housing etc. 

Also for your point 1: Black, Latino and NA more likely to hold jobs that have to be done at their workplace rather than remotely…I am not sure this is totally true, it’s jobs that can’t be done from home which could mean they are working during the pandemic or could mean they have lost their job altogether (See Baker 2020 in AJPH: https://ajph.aphapublications.org/doi/full/10.2105/AJPH.2020.305738)

As we try to make clearer in the introduction, the goal of the paper is to describe potential workplace risks of exposure to COVID-19 by race and ethnicity – in order to assess whether workplace conditions are likely to play an important role in the higher rates of COVID-19 infection among Black, Latino, Native American, and some other groups of workers. We agree that access to healthcare and the ability to take time off from work, as well as living in crowded households, living in neighborhoods with more infected or fewer vaccinated people, etc., affect the probability that workers bring SARS-CoV-2 into their workplace and therefore, expose co-workers to the virus. However, accounting for the multiple reasons that workers in disadvantaged groups may be more likely to contract COVID-19 is beyond the scope of this analysis.

As we noted above, we use occupational standing in the analysis, not as a measure of an individual’s socioeconomic status, but as a proxy variable for the likelihood that workers in each occupation have access to risk mitigation measures in their workplace. We describe the logic of doing so both in the paper and in response to Reviewer #3’s comments. Occupational standing is based on the average value of those employed in the occupation rather than on an individual measure – thus, it is an occupation-level, not an individual-level, variable. We have revised the section in the paper on “Occupational Standing” to explain why we chose occupational education over occupation income:

We chose occupational education rather than occupational income – another well-established indicator of occupational ranking – both because educational attainment is typically reported with greater accuracy than income or wages and because among those with a valid occupational code in the ACS data file provided by IPUMS, educational attainment information is complete while information on income is missing for one-fifth of individuals. The correlation between these two measures of occupational ranking across the occupations in the dataset is high (0.72).

As we also note in the paper, our analytic procedure does not permit inclusion of those who lost their job, a loss that varies by race and ethnicity. To further emphasize this issue, we have added a sentence emphasizing your point above (in the Analytic Strategy section) and citing the Baker paper.

-Line 66: “we investigate these claims”—previously you cited peer-reviewed literature, seems reasonable to refer to that as more than a claim. Maybe “we investigate these hypotheses” or “these findings”. Claims makes it seems like you are telling occupational health researchers they are just making up things.

This is a matter of word usage in different disciplines. We have changed ‘claims” to “issues” in the Abstract and we have rewritten the relevant sentence in the Introduction.

-Is there a reason you aren’t including median pay in your analysis? That is very easy to access and would give more weight to the “occupational standing”—and it’s easy to integrate with O*NET data

See our comment above with regard to why we prefer occupational education to occupational income.

-Line 93: “lower status workers”—is this really want you mean? It’s only higher status because we as educated people have decided what is higher v. lower status work. Status really more an opinion than something that can be measured. Maybe “lower paid”, “more vulnerable”, “precarious” “underrepresented” ??

Occupational status or standing is a sociological term representing objective location (as defined by a specific variable or set of variables) in the social hierarchy. The field of sociology has an extensive literature on how to measure occupational status or standing consistently and reliably. As we noted in response to Reviewer #3, who had similar concerns, we no longer use terms such as lower status or higher status workers and occupations. Instead, we refer to occupational standing, or OS for clarity, which is defined clearly in the text, in terms of an average educational level for all workers in that occupation.

-Line 95: yes, underrepresented workers in more precarious jobs often have less access to risk mitigation strategies and workplace POWER than those in higher paid job. Worth mentioning the power element here. Maybe no one is getting proper PPE but there are some workers who could more easily ask than others

We agree and have modified the text. We also note in the Discussion section that workers in high OS occupations are more likely “to demand risk reduction measures and have the bargaining power to obtain them” (1st paragraph) and later in the Discussion section that workers in low OS occupations have “less control and decision-making ability over how their workplaces are run.”

Data

-Can you defend why you used 1 year ACS data instead of 5 year? Especially for small demographic groups, 1 year is extremely noisy—5 year would be better. When you start stratifying by groups you start to lose reliability of the estimate with only 1 year ACS data.

See: https://www.census.gov/content/dam/Census/library/publications/2018/acs/acs_general_handbook_2018_ch03.pdf

Yes, we can defend this decision. We now note in the first sentence of the Data section that occupation is based on a response to a question on the current or most recent job or business held in the five years prior to the ACS. We also note (in the Analytic Strategy section) that 88% of the responses to this question pertain to jobs or businesses held in the one year prior to the ACS. If we were to use the 5-year ACS, many of the jobs in our analysis would not be very recent ones (e.g., they could go back as far as a decade, which would include, for example, the Great Recession), making this information less relevant to the period of the pandemic. We emphasize in the Discussion section the importance of capturing jobs close to the pandemic period. 

The ACS samples a large number of people each year. In our data set we have about 1.9 million respondents who reported jobs. As you can see from Table 1, the smallest groups have about 3,000 recent workers each and only three groups have below 10,000 recent workers; there are over 160,000 Black recent workers and over 250,000 Latino recent workers. These sample sizes are much larger than those in almost every other survey and sufficiently large to stratify by quartiles of occupational standing. 

-Worth noting O*NET isn’t just a sample of workers—also includes employers and job experts. (line 114)

Yes, you are correct. We have moved the reference to workers to the subsequent sentence to clarify that the data we use on hazardous job characteristics come from incumbent workers.

-Line 120: virus is primarily transmitted through aerosols. This should at least be listed first (before respiratory droplets).

We have reversed the order of aerosols and respiratory droplets.

Results

-I’m quite confused about the numbers in your sample set. There are 160+ million workers in the USA. How come you only have so few? Given many of the groups that you report (Chinese, Filipino, Vietnamese, etc.) have less than 65,000 people, the 1 year ACS data is likely not very valuable and you should be using 5 year data instead to have more valid estimates.

Altogether, our sample includes nearly 2 million workers. As we noted earlier, for this analysis we chose to prioritize the recency of the data, leading to our use of the 1-year ACS. The guidance from the Census Bureau that you reference regarding the use of the 5-year ACS for populations below 65,000 refers to geographic areas rather than racial/ethnic subgroups.

See: https://www.census.gov/programs-surveys/acs/guidance/estimates.html

-Line 251-253: this type of commentary seems more like discussion section fodder

We reworded this sentence.

Your results assume that the (rudimentary) exposure metrics you included capture all occupational exposure risk with statements like “…Black and Latino workers do not face higher occupational risks related to viral exposure compared with Whites.” Instead what you mean to say is …”Black and Latino workers are less likely to be employed in occupations with characteristics related to increased exposure to SARS-CoV-2, as ascertained from five O*NET metrics.” Your clarifying sentence after this is also helpful and welcomed! However, do I think more of this should be in the discussion as opposed to in the results.

We toned down the sentence regarding Black and Latino workers to make the result more tentative.

-Line 279: And be paid a living wage, have access to a union, etc.

We agree that there are many other key differences between physicians and home health workers, including pay, union access and others. However, our point here was that workers in two occupations with a high level of potential exposure to the coronavirus through contact with infected or potentially infected people may differ considerably in access to PPE and other risk mitigation measures. 

Discussion

-I struggle (as another reviewer did) with saying that years of college education is a reasonable proxy for “occupational standing”. I think at a minimum wage should also be considered. This to me is a very “white collar” interpretation of occupational standing. But in the trades, where workers enjoy high pay, strong union protections, and access to benefits/safety…most have not completed a year of college. Pay/union representation seem equally important to me. Instead of using just the one variable as a proxy for “occupational standing” instead just refer to it as “educational attainment” with an understanding that we are imposing very white collar values on what makes “good work” and “important work”

As we note above, occupational education is not the same as individual educational attainment. Instead, it is an average of the educational attainment (percent with some college education) of all workers in a particular occupation. Occupational education and occupational income are two standard sociological indicators of location within the social hierarchy – standing which empirically provides access to power and control. We agree that union representation can be very important, but we don’t have any information on this issue. 

See our earlier response as to why we use occupational education rather than occupational income. Use of both variables would be problematic since they are highly correlated. We include this information in the revised text.

-Line 375: “higher status”—suggest eliminating this phrasing as detailed in a previous comment; Line 383—another stray “status”

Done.

-Line 377: this gets at some of what I was detailing earlier, but I don’t think this necessarily relates to years of college. The trades/all goods producing occupations would fall into these categories even if their workers didn’t go to college (unless trade school is counted as college, but that seems unlikely)

Again, keep in mind that occupational education refers to an average educational attainment for all workers in a particular occupation, not for an individual. We agree that for some subgroups of occupations, the occupational standing variable (the percent of workers within each occupation who have completed at least a year of college) may not do as good a job as other variables at predicting which workers have more or less access to COVID-19 mitigation measures. Obviously, it would be ideal to have a direct measure of use of mitigation measures in each workplace. Since such a direct measure is not available and we are comparing across a very wide range of occupations, we believe that occupational standing, defined here as occupational education, does a reasonable job as a proxy for occupations in which workers are more likely to have the power and control necessary to insure COVID-19 mitigation measures in their workplace. As noted above, occupational education is highly correlated with occupational income and so substituting occupational income as the proxy would not greatly change our results. 

-Line 409: I get that there are differences between “essential” and “frontline” (very militaristic way of referring to these workers!) but wouldn’t the same limitations apply? “Frontline” would also have variability by states and over time? For example, if a restaurant is closed then those are not “frontline” workers, but if open then they would be, wouldn’t they?

We have adopted the language (essential and frontline) used in studies of work exposure during the pandemic. Our point is that the definition of “essential” workers – which was defined by federal and state agencies – has changed frequently over the course of the pandemic and across space. In contrast, the definition of “frontline” we use is static – it is defined based on an assessment (using O*NET variables) of whether or not jobs could be done at home. It seems much less likely that jobs would change rapidly or frequently from being able to be done at home to not being so. You are correct that there is variability in terms of which workers are frontline primarily because many frontline workers lost their jobs due to closure (although some reductions in frontline exposure may be temporary). We now emphasize this issue of job loss in the second paragraph of the Analytic Strategy. We also mention the issue of job loss when we describe limitations of our analysis in the Discussion section. Nevertheless, frontline worker status is more straightforward to define and has resulted in a more consistent classification across studies than essential worker status. 

Limitations

-Limitations with using 1 year ACS v. 5 year ACS especially for the small groups you are considering?

-Limitations because using the ACS means you have to combine some detailed occupations so you miss out on variability between occupations?

-Only able to look at a small subset not the whole workforce

-O*NET limitations: doesn’t cover all occupations

We explained above why use of the 1-year ACS is preferable to the 5-year ACS. The ACS includes a very large high-quality unbiased random sample of the entire workforce. Moreover, O*NET does cover almost all occupations: we are able to link almost all occupations with O*NET (an exception is the military; we have added information on the number of ACS respondents in the military). In response to your concern, we added a sentence in the Discussion regarding the limitation of having to combine detailed occupations that may reflect different risk exposures.

-O*NET limitations: subject to misclassification and undercounting, generated from subjective questionnaires, differential misclassification across groups (e.g. folks may not know they are exposed to disease/infection unless they are told they are and those risks are communicated making people in healthcare more likely to realize than other occupations), within-job exposure variation is not accounted for

We have added sentences regarding response error in the O*NET surveys (as well as the ACS). At the end of the paragraph, we address your concern regarding within-job exposure variation (we note such variation by age, sex, and race/ethnicity of workers).

-General limitations of your study: Many, many occupations/people are missing (some of which would be considered “frontline” and/or lower standing): independent contractors, domestic workers, military, is agriculture included?, self-employed, contingent, undocumented…These are also jobs that are more diverse

ACS uses a population-based sample that includes respondents regardless of type of work or other characteristics such as documentation status. Particular types of jobs are missing from our analysis only if none of the nearly 2 million respondents in the ACS reported holding that job (in the previous five years). In these cases, the jobs were simply not held during this period rather than being technically “missing.” The sample includes independent contractors, domestic workers, military personnel, agricultural workers, self-employed workers and those in contingent employment. Although there may be higher non-response rates among undocumented workers, ACS data have been used to estimate the size and characteristics of the undocumented population specifically because adults are included regardless of documentation status (Capps et al., 2018). As noted above, with the exception of military workers who are not included in O*NET, we are able to match virtually all occupations held by ACS workers with O*NET job characteristics. We agree that there is diversity in activities and exposure within jobs, but there is not much we can do about it.

Note that the occupations listed in S2 Appendix include only the most frequently held occupations in each OS quartile (and by risk factor). Thus, many occupations we include in our analysis are not listed in this appendix.

Capps R, Bachmeier JD, Van Hook J. Estimating the Characteristics of Unauthorized Immigrants Using U.S. Census Data: Combined Sample Multiple Imputation. The ANNALS of the American Academy of Political and Social Science. 2018;677: 165–179. doi:10.1177/0002716218767383

-General limitation: don’t include wages which is likely a larger predictor for how a pandemic would affect a worker

As we describe above, we are not attempting to predict how the pandemic would affect individual workers.

-Line 468: Disagree. If employers reduce workplace transmission, then community transmission is also reduced. We have to stop separating “occupational health” and “public health”. They are interconnected and if we protect workers, we are positively benefitting the community.

We completely agree that workplace transmission is a part of community transmission and have rewritten that sentence in the Discussion section to acknowledge this interdependence. 

-Line 470: Thank you for giving some examples! However, these examples really only work for well paid, likely white workers. “preventing workers with symptoms or positive tests from coming to work” Does little for the Black taxi driver who has to make money in order to feed his family and taking a week off work could financially devastate him. How about instead, “providing paid sick leave that is ample and accessible, for which workers do not face retaliation or retribution for taking, if they have an exposure event” Or something like that.

We have included your example of sick leave in our revised discussion.

-Line 475: is this still true about testing? I think we are at a point where we have a range of rapid and accurate tests…and isn’t all testing free even to the uninsured? People might not realize that, but it is. 

We have added a sentence in the Introduction to indicate that our results best reflect the situation at the end of 2020, although they continue to be relevant.

-Line 486: Exposure controls are all the controls you are mentioning? Delete exposure. Also, “employers can provide PPE?” How about employers must or employers should

In the previous version, we describe examples of the full range of controls outlined by Carlsten et al. (citation in the paper references) based on industrial hygiene’s hierarchy of controls model. We did so at the explicit request of a PLOS ONE reviewer for the last round of reviews of this paper. At the request of Reviewer #3 for this round of the paper, we have now summarized and shortened this section.

Figure 1

-Very blurry. Would be helpful to have the x-axis on the male section as well, it can be hard to follow it all the way down to the bottom

We suspect the Figure became less clear when we ran it through the required software. We have now created a higher resolution figure and added the racial/ethnic groups to the male graph.

---

## [Decision Letter · Decision Letter 2]

22 Jul 2021

PONE-D-20-36345R2

Racial and Ethnic DIfferentials in COVID-19-Related Job Exposures by Occupational Standing in the US

PLOS ONE

Dear Dr. Goldman,

Thank you for submitting your manuscript to PLOS ONE. After careful consideration, we feel that it has merit but does not fully meet PLOS ONE’s publication criteria as it currently stands. Therefore, we invite you to submit a revised version of the manuscript that addresses the points raised during the review process.

ACADEMIC EDITOR: 

Please note that I have requested 2 minor but important edits to acknowledge the limitations of the dataset (with respect to workers that may be excluded or misclassified) and the descriptive nature of the study (to temper interpretations around causality). Once these edits have been made, I will accept the manuscript. No other changes are required. Please submit your revised manuscript within 30 days of receipt of this email. If you will need more time than this to complete your revisions, please reply to this message or contact the journal office at plosone@plos.org. Please include the following items when submitting your revised manuscript:
A rebuttal letter that responds to each point raised by the academic editor and reviewer(s). You should upload this letter as a separate file labeled 'Response to Reviewers'.A marked-up copy of your manuscript that highlights changes made to the original version. You should upload this as a separate file labeled 'Revised Manuscript with Track Changes'.An unmarked version of your revised paper without tracked changes. You should upload this as a separate file labeled 'Manuscript'.

We look forward to receiving your revised manuscript.

Kind regards,

Marlene Camacho-Rivera, ScD, MPH

Academic Editor

PLOS ONE

Journal Requirements:

Additional Editor Comments (if provided):

I appreciate the considerable effort that has gone into revising this manuscript twice and have forwarded this decision without the additional review. There are 2 very minor edits, which can be resolved quickly, but are important given the potential contributions of the paper:

1. Please make more explicit the limitations of the O*NET sampling frame and mention the types of workers that are left out of the O*NET sampling frame.

2. Please make more explicit the descriptive nature of the study, both in the study objectives and limitations section.

With these 2 edits, the manuscript will be accepted.

Reviewers' comments:

Reviewer's Responses to Questions

**Comments to the Author**

1. If the authors have adequately addressed your comments raised in a previous round of review and you feel that this manuscript is now acceptable for publication, you may indicate that here to bypass the “Comments to the Author” section, enter your conflict of interest statement in the “Confidential to Editor” section, and submit your "Accept" recommendation.

Reviewer #3: All comments have been addressed

Reviewer #4: (No Response)

2. Is the manuscript technically sound, and do the data support the conclusions?

Reviewer #3: Partly

Reviewer #4: Yes

3. Has the statistical analysis been performed appropriately and rigorously? 

Reviewer #3: I Don't Know

Reviewer #4: Yes

4. Have the authors made all data underlying the findings in their manuscript fully available?

Reviewer #3: Yes

Reviewer #4: No

5. Is the manuscript presented in an intelligible fashion and written in standard English?

Reviewer #3: Yes

Reviewer #4: Yes

6. Review Comments to the Author

Reviewer #3: The authors have addressed the majority of the comments/aspects raised by the reviewers. However, even if both reviewers raised concerns about a limitations section, the authors did not incorporate it. Although, authors included some sentences throughout the discussion about the concerns mentioned and the manuscript has been greatly improved, there are some issues that remain and are worthy of clarification. For instance, the issues of confounding and heterogeneity raised still hold. For instance, the response indicates that the study is only descriptive. However, this has not been stated anywhere in the text and the objective of "extending previous work on racial/ethnic differences in potential work-related exposure to COVID-19 by examining 12 racial/ethnic groups and by considering five indicators of potential risk exposure" including to "examine potential risk of exposure separately by occupational standing (OS)," may not be understood as only descriptive. Given the use of "control for OS", as stated by the authors in their answer. If the authors did not intended the estimation a causal effect and association, nor the interpretation as such, this could be explicitly presented in the abstract, objective/methods and discussion section of the manuscript as well, so the interpretations could be accordingly made.

Reviewer #4: Thanks to the authors for the improvements to the paper, both minor and major.

I have just a few lingering concerns after which adopted, I will consider the paper to be publication ready.

A few places in your comments to reviewer you emphasize that O*NET covers all occupations, and that the ACS survey captures all occupations. I think the fallacy you are falling into is the assumption that the ~700 SOC codes covers all workers in the United States which is just not the case (you do acknowledge that military is not included in O*NET). Given the way O*NET is collected (where they reach out to workers/employers/experts about each of the occupations covered by SOCs), many workers are left out of the data collection—for example, Uber drivers wouldn’t be surveyed by O*NET, or a delivery worker for Amazon, a temporary worker for any organization, or a domestic worker being paid under the table, or even a general contractor, or a migrant agricultural worker. These workers may be surveyed by ACS, but their reported occupation is being matched to a census occupation code that isn’t very precise, and then you are further matching to O*NET data that was generated to represent a different population of workers (i.e. only those in more traditional employer-employee relationships covered by a SOC code). Employee respondents to O*NET are found through their employer. The authors acknowledge there is inner-occupation variability, but “there is not much we can do about it.” I am only asking for you to acknowledge these limitations—(1)collapsing all the diverse occupations self-reported in the ACS data to 400 occupations induces inner-occupation variability, and (2) in matching ACS to O*NET data, you are comparing two different populations (ACS is general US cross section, O*NET is a very specific set of occupations marked by a typical employee-employer relationship and is only available in English and Spanish) that have different distributions of occupational variability.

Neither of these limitations negate your work, but it has to be mentioned that O*NET does not cover all workers and indeed many of the workers it does not cover may be more “frontline” or at increased risk (and also could be more likely to be workers of color). By not acknowledging these “missing” workers that are continuously left out of BLS data collection it is further perpetuating their decreased standing and lack of importance and necessity to our economy.

Moreover, the lack of data from O*NET on the demographics of respondents makes it hard to ensure the sample is representative of the American workforce, though most assume that it is.

I would suggest reviewing this paper if you haven’t already: https://onlinelibrary.wiley.com/doi/full/10.1002/ajim.20846?casa_token=MnvFp1NvrwUAAAAA%3ACffACaImtyoCu67O96UX_SBe49mfnSxZWlzrI8Rzkr9xmbeuL0uv3kVWI16Xb8geBNN2JnpXV_9n1Jg

Thank you for helping me understand the use of 1 year v. 5 year ACS. I agree that in this analysis the currency of the data is preferable to the precision of the data.

7. PLOS authors have the option to publish the peer review history of their article (what does this mean?). If published, this will include your full peer review and any attached files.

Reviewer #3: No

Reviewer #4: No

---

## [Author Response · Author response to Decision Letter 2]

26 Jul 2021

We thank the reviewers for their helpful comments. Below we provide responses to each of the reviewer’s comments. The reviewer’s comments are presented in italics.

Reviewer 3:

The authors have addressed the majority of the comments/aspects raised by the reviewers. However, even if both reviewers raised concerns about a limitations section, the authors did not incorporate it. Although, authors included some sentences throughout the discussion about the concerns mentioned and the manuscript has been greatly improved, there are some issues that remain and are worthy of clarification. For instance, the issues of confounding and heterogeneity raised still hold. For instance, the response indicates that the study is only descriptive. However, this has not been stated anywhere in the text and the objective of "extending previous work on racial/ethnic differences in potential work-related exposure to COVID-19 by examining 12 racial/ethnic groups and by considering five indicators of potential risk exposure" including to "examine potential risk of exposure separately by occupational standing (OS)," may not be understood as only descriptive. Given the use of "control for OS", as stated by the authors in their answer. If the authors did not intended the estimation a causal effect and association, nor the interpretation as such, this could be explicitly presented in the abstract, objective/methods and discussion section of the manuscript as well, so the interpretations could be accordingly made.

As noted previously, we did not intend this paper to provide a causal analysis. As suggested by the reviewer, we now explicitly state that the analysis is descriptive. We have added wording to the Abstract, Introduction, Analytic Strategy and Discussion sections to make this very clear. For example, in the Discussion we now say the following: “The analysis is descriptive – portraying potential workplace risks by race, ethnicity and gender – rather than an investigation of the root causes of these patterns.”

Reviewer #4: 

Thanks to the authors for the improvements to the paper, both minor and major.

I have just a few lingering concerns after which adopted, I will consider the paper to be publication ready.

A few places in your comments to reviewer you emphasize that O*NET covers all occupations, and that the ACS survey captures all occupations. I think the fallacy you are falling into is the assumption that the ~700 SOC codes covers all workers in the United States which is just not the case (you do acknowledge that military is not included in O*NET). Given the way O*NET is collected (where they reach out to workers/employers/experts about each of the occupations covered by SOCs), many workers are left out of the data collection—for example, Uber drivers wouldn’t be surveyed by O*NET, or a delivery worker for Amazon, a temporary worker for any organization, or a domestic worker being paid under the table, or even a general contractor, or a migrant agricultural worker. These workers may be surveyed by ACS, but their reported occupation is being matched to a census occupation code that isn’t very precise, and then you are further matching to O*NET data that was generated to represent a different population of workers (i.e. only those in more traditional employer-employee relationships covered by a SOC code). Employee respondents to O*NET are found through their employer. The authors acknowledge there is inner-occupation variability, but “there is not much we can do about it.” I am only asking for you to acknowledge these limitations—(1)collapsing all the diverse occupations self-reported in the ACS data to 400 occupations induces inner-occupation variability, and (2) in matching ACS to O*NET data, you are comparing two different populations (ACS is general US cross section, O*NET is a very specific set of occupations marked by a typical employee-employer relationship and is only available in English and Spanish) that have different distributions of occupational variability.

Neither of these limitations negate your work, but it has to be mentioned that O*NET does not cover all workers and indeed many of the workers it does not cover may be more “frontline” or at increased risk (and also could be more likely to be workers of color). By not acknowledging these “missing” workers that are continuously left out of BLS data collection it is further perpetuating their decreased standing and lack of importance and necessity to our economy.

Moreover, the lack of data from O*NET on the demographics of respondents makes it hard to ensure the sample is representative of the American workforce, though most assume that it is.

I would suggest reviewing this paper if you haven’t already: https://onlinelibrary.wiley.com/doi/full/10.1002/ajim.20846?casa_token=MnvFp1NvrwUAAAAA%3ACffACaImtyoCu67O96UX_SBe49mfnSxZWlzrI8Rzkr9xmbeuL0uv3kVWI16Xb8geBNN2JnpXV_9n1Jg

Thank you for helping me understand the use of 1 year v. 5 year ACS. I agree that in this analysis the currency of the data is preferable to the precision of the data.

We have changed the text in several places in the Discussion to make these points. We acknowledge that the ACS classification scheme has resulted in the combining of O*NET occupations thereby increasing intra-occupation variability. In a subsequent paragraph, we discuss the likely consequence of O*NET surveys being restricted to formal workers – i.e., underestimation of the extent of racial and ethnic differences in workplace exposures to COVID-19. We acknowledge that we are probably capturing substantial numbers of informal workers in the ACS data.

---

## [Decision Letter · Decision Letter 3]

30 Jul 2021

Racial and Ethnic DIfferentials in COVID-19-Related Job Exposures by Occupational Standing in the US

PONE-D-20-36345R3

Dear Dr. Goldman

We’re pleased to inform you that your manuscript has been judged scientifically suitable for publication and will be formally accepted for publication once it meets all outstanding technical requirements.

Kind regards,

Marlene Camacho-Rivera, ScD, MPH

Academic Editor

PLOS ONE

Additional Editor Comments (optional):

Reviewers' comments:

Reviewer's Responses to Questions

**Comments to the Author**

1. If the authors have adequately addressed your comments raised in a previous round of review and you feel that this manuscript is now acceptable for publication, you may indicate that here to bypass the “Comments to the Author” section, enter your conflict of interest statement in the “Confidential to Editor” section, and submit your "Accept" recommendation.

Reviewer #4: All comments have been addressed

2. Is the manuscript technically sound, and do the data support the conclusions?

Reviewer #4: (No Response)

3. Has the statistical analysis been performed appropriately and rigorously? 

Reviewer #4: (No Response)

4. Have the authors made all data underlying the findings in their manuscript fully available?

Reviewer #4: (No Response)

5. Is the manuscript presented in an intelligible fashion and written in standard English?

Reviewer #4: (No Response)

6. Review Comments to the Author

Reviewer #4: (No Response)

7. PLOS authors have the option to publish the peer review history of their article (what does this mean?). If published, this will include your full peer review and any attached files.

Reviewer #4: No

---

## [Editor Report · Acceptance letter]

6 Aug 2021

PONE-D-20-36345R3 

Racial and ethnic differentials in COVID-19-related job exposures by occupational standing in the US 

Dear Dr. Goldman:

I'm pleased to inform you that your manuscript has been deemed suitable for publication in PLOS ONE. Congratulations! Your manuscript is now with our production department. 

Kind regards, 

on behalf of

Dr. Marlene Camacho-Rivera 

Academic Editor

PLOS ONE